# Finite-Time Analysis of Discrete-Time Stochastic Interpolants

**Yuhao Liu** [1]   **Yu Chen** [1]   **Rui Hu** [1]   **Longbo Huang** [1]

## Abstract

The stochastic interpolant framework offers a powerful approach for constructing generative models based on ordinary differential equations (ODEs) or stochastic differential equations (SDEs) to transform arbitrary data distributions. However, prior analyses of this framework have primarily focused on the continuous-time setting, assuming a perfect solution of the underlying equations. In this work, we present the first discrete-time analysis of the stochastic interpolant framework, where we introduce an innovative discrete-time sampler and derive a finite-time upper bound on its distribution estimation error. Our result provides a novel quantification of how different factors, including the distance between source and target distributions and estimation accuracy, affect the convergence rate and also offers a new principled way to design efficient schedules for convergence acceleration. Finally, numerical experiments are conducted on the discrete-time sampler to corroborate our theoretical findings.

## 1. Introduction

Stochastic interpolants (Albergo & Vanden-Eijnden, 2023; Albergo et al., 2023) provide a general framework for constructing continuous mappings between arbitrary distributions. This framework draws inspiration from flow-based and diffusion-based models, which generate samples by continuously transforming data points from a base distribution to a target distribution via learned ordinary differential equations (ODEs) or stochastic differential equations (SDEs).

Within the stochastic interpolant framework, one obtains learnable ODEs or SDEs that transport data by defining an interpolation between data points sampled from different distributions. This framework offers significant design flexibility and has demonstrated promising results in various

applications, including probabilistic forecasting (Chen et al., 2024b), image generation (Ma et al., 2024; Albergo et al., 2024), and sequential modeling (Chen et al., 2024a).

Despite its potential in real-world applications, there remains a gap between the theoretical analyses and practical implementations of stochastic interpolants. In practical scenarios, instead of perfectly solving the underlying equations, one can only access a learned estimator for a finite number of time steps, which necessitates the use of discrete-time samplers to simulate the true continuous generation process. However, previous analyses have largely focused on continuous-time generation, assuming perfect solutions to the underlying equations. This leads to a crucial question for bridging the gap:

> **What is the convergence rate of discrete-time stochastic interpolant, and how to enhance its performance algorithmically?**

Similar problems have been studied in the theories of diffusion models, and most results were derived based on Girsanov-based methods in SDE analyses, which reduce the problem to providing upper bounds on the discretization errors. Specifically, existing analyses on the discretization errors can be mainly categorized into two types. The first type partitions the error into space-discretization and time-discretization. Among them, Lee et al. (2022) and Chen et al. (2023d) assume a uniform Lipschitz constant on the score function, while Chen et al. (2023a) do not, but they all utilize the Markovian property and the Gaussian form of the diffusion process to obtain their results. The second type uses Itô's calculus to obtain upper bounds, such as Benton et al. (2024), who adapt existing results from stochastic localization to produce tight bounds by finding the equivalence between two methods.

However, the aforementioned ideas do not apply to stochastic interpolants due to the following key difference: the stochastic interpolant framework in consideration has a distinct structure introduced by a random interpolation between two distributions instead of a linear combination of one distribution with Gaussian, destroying the Markovian property. This difference not only necessitates novel analysis for the discretization errors of score functions but also requires additional analysis to bound the discretization errors for the

---

[1]IIIS, Tsinghua University, Beijing, China. Correspondence to: Longbo Huang <longbohuang@tsinghua.edu.cn>.

*Proceedings of the 42nd International Conference on Machine Learning*, Vancouver, Canada. PMLR 267, 2025. Copyright 2025 by the author(s).

velocity function, which arises from the general interpolation function introduced in this context.

To tackle the above challenges, in this work, we offer the first finite-time convergence bound in Kullback-Leibler (KL) error for the SDE-based generative model within the stochastic interpolant framework. Our result presents a novel analysis building on existing Girsanov-based techniques. In the analysis of discretization errors, one key highlight of our approach is modeling the evolution of discretized terms via stochastic calculus. This allows us to decompose the discretization error into components linked to derivatives of conditional expectation. Notably, we leverage the Gaussian latent variables embedded within our stochastic interpolants, enabling the explicit representation of these derivatives as conditional expectations, and hence providing a key solution to the challenges.

**Our contributions.** The main contributions of our paper are as follows.

(i) This work presents the first finite-time convergence bound for the SDE-based generative model within the stochastic interpolant framework. Specifically, we formulate the discrete-time sampler using the Euler–Maruyama scheme and derive a general error bound on the generative process, which notably does not require Lipschitz assumptions on the score functions or velocity functions. This setting aligns with recent analyses of score-based diffusion models that relax Lipschitz assumptions on the score functions, such as those by Chen et al. (2023a) and Benton et al. (2024).

(ii) We propose a novel schedule of step sizes and rigorously bound the number of steps required to achieve an $\varepsilon^2$ KL-error. In the specific case where the base distribution $\rho_0$ is Gaussian, where our setting reduces to the standard diffusion model, our bound achieves the same order of dependence as that by Chen et al. (2023a).

(iii) We implement the sampler with our proposed schedule and conduct a comparison against using uniform step sizes. Our results validate the theoretical findings and demonstrate the superior performance of our schedule when no additional regularity conditions are assumed.

## 2. Related Works

**Stochastic Interpolants** The concept of stochastic interpolants is introduced by Albergo & Vanden-Eijnden (2023), establishing a framework for constructing generative models based on continuous-time normalizing flows. Building upon this foundation, Albergo et al. (2023) extend the framework by incorporating Gaussian noise into the interpolant, effectively unifying flow-based and diffusion-based methods. Both Albergo & Vanden-Eijnden (2023) and Albergo et al.

(2023) investigate the impact of using estimators instead of the true velocities in the equations.

Following the stochastic interpolants framework, several works have focused on specific applications. Albergo et al. (2024) utilize the framework to develop novel data coupling methods, addressing image generation tasks such as in-painting and super-resolution. Chen et al. (2024b) and Chen et al. (2024a) adapt the conditional generation framework with stochastic interpolants to tackle future state prediction and sequential modeling problems, respectively.

**Convergence Analysis of Diffusion Models** Numerous results have been established on the convergence rates of diffusion models under various data assumptions (Bortoli, 2022; Lee et al., 2022). Notably, Chen et al. (2023d) employ an approximation argument to apply Girsanov's theorem in scenarios where the Novikov condition does not hold. Based on this analysis, Chen et al. (2023d) and Chen et al. (2023a) provide error bounds assuming Lipschitz score functions. Chen et al. (2023a) also develop a KL error bound without requiring Lipschitzness assumptions, leveraging early stopping. This bound is subsequently improved by Benton et al. (2024), achieving a $\varepsilon^2$ KL error in $\tilde{O}\left(\frac{d\log^2(1/\delta)}{\varepsilon^2}\right)$ steps, which exhibits near-linear dependence on the dimension $d$. In addition to convergence rate analysis, several works have focused on problems such as score approximation (Chen et al., 2023b), improved DDPM samplers (Liang et al., 2024; Li et al., 2024b;a) and ODE-based methods (Chen et al., 2023c).

## 3. Preliminaries: Stochastic Interpolants

In this paper, we consider continuous-time stochastic processes that bridge any two arbitrary probability distributions in finite time. Formally, given two probability distributions $\rho_0$ and $\rho_1$ in $\mathbb{R}^d$, a stochastic interpolant (Albergo et al., 2023; Albergo & Vanden-Eijnden, 2023; Albergo et al., 2024) is a stochastic process defined as:

$$x_t = I(t, x_0, x_1) + \gamma(t)z, \qquad t \in [0, 1] \qquad (1)$$

where $I$ is a twice-continuously differentiable interpolation function satisfying the boundary conditions $I(0, x_0, x_1) = x_0$ and $I(1, x_0, x_1) = x_1$, and there exists a constant $C > 0$ such that for all $t \in [0, 1]$ and $x_0, x_1 \in \mathbb{R}^d$,

$$\partial_t I(t, x_0, x_1) \leq C\|x_0 - x_1\|. \qquad (2)$$

Here $\gamma(t)$ is a time-dependent scale function with $\gamma(t)^2 \in C^2[0, 1]$, $\gamma(0) = \gamma(1) = 0$, and $\gamma(t) > 0$ for $t \in [0, 1]$. This definition indicates that $\left|\frac{d}{dt}(\gamma^2(t))\right|$ is bounded by a constant. $(x_0, x_1)$ is drawn from a joint measure $\nu$ with marginals $\rho_0$ and $\rho_1$, i.e., $\rho_0(dx_0) = \nu(dx_0, \mathbb{R}^d)$ and $\rho_1(dx_1) = \nu(\mathbb{R}^d, dx_1)$. $z \sim \mathcal{N}(0, I_d)$ is a standard Gaussian variable independent of $(x_0, x_1)$.

In the definition (1), $I(t, x_0, x_1)$ represents the interpolation component, while $\gamma(t)z$ introduces a Gaussian latent term crucial for subsequent analysis. We denote the density of $x_t$ by $\rho(t, \cdot)$ or simply $\rho(t)$. According to the construction, the stochastic interpolant satisfies $\rho(0) = \rho_0$ and $\rho(1) = \rho_1$. This framework allows for a wide range of interpolation functions $I(t, x_0, x_1)$ and scale functions $\gamma(t)$, offering significant flexibility in design.

Stochastic interpolants provide a framework for generative modeling through stochastic differential equations. As shown by Albergo et al. (2023), when $\mathbb{E}_{(x_0,x_1)\sim\nu}\|\partial_t I(t, x_0, x_1)\|^4$ and $\mathbb{E}_{(x_0,x_1)\sim\nu}\|\partial_t^2 I\|^2$ are bounded, the solution to the following forward SDE

$$dX_t^F = b_F(t, X_t^F)dt + \sqrt{2\epsilon(t)}dW_t, X_0 \sim \rho_0 \quad (3)$$

satisfies $X_t^F \sim \rho(t)$ for all $t \in [0, 1]$. Here $\epsilon \in C[0, 1]$ is a non-negative function, and the drift term $b_F$ is defined as

$$
\begin{aligned}
b_F(t, x) &= b(t, x) + \epsilon(t)s(t, x), \\
b(t, x) &= \mathbb{E}[\dot{x}_t | x_t = x] \\
&= \mathbb{E}[\partial_t I(t, x_0, x_1) + \dot{\gamma}(t)z | x_t = x], \\
s(t, x) &= \nabla \log \rho(t, x) = -\gamma^{-1}(t)\mathbb{E}[z | x_t = x].
\end{aligned}
\quad (4)
$$

In the definition (4), $s(t, x)$ is the well-known score function, and $b(t, x)$ represents the mean velocity field of (1) (following the notations of Albergo et al. 2023). This implies that a process starting from $\rho_0$ and evolving according to the forward SDE (3) will have density $\rho(t)$ at time $t$. Consequently, at time $t = 1$, the process will have the desired target density $\rho_1$. This establishes a stochastic mapping from $\rho_0$ to $\rho_1$, providing a foundation for generative modeling.

Following Albergo et al. (2023), we further introduce the velocity function $v(t, x)$ as:

$$
\begin{aligned}
v(t, x) &= \mathbb{E}[\partial_t I(t, x_0, x_1) | x_t = x] \\
&= b(t, x) + \dot{\gamma}\gamma s(t, x).
\end{aligned}
\quad (5)
$$

Both $b(t, x) = v(t, x) - \dot{\gamma}(t)\gamma(t)s(t, x)$ and $b_F(t, x) = b(t, x) + \epsilon(t)s(t, x)$ are linear combinations of $v(t, x)$ and $s(t, x)$. The function $\epsilon(t)$ controls the level of randomness in the mapping from $\rho_0$ to $\rho_1$. When $\epsilon(t) \equiv 0$ the SDE reduces to an ODE. We assume $\epsilon(t) = \epsilon$ is a constant without loss of generality, similar to Albergo et al. (2023) and dos Santos Costa et al. (2024).

**Connection with Diffusion Models.** Consider the special case where $x_0$ and $x_1$ are independent, with $x_0 \sim \mathcal{N}(0, I_d)$. Let $I(t, x_0, x_1) = (1 - t)x_0 + tx_1$ and $\gamma(t) = \sqrt{2t(1 - t)}$. Then, the stochastic interpolant can be expressed as

$$x_t = tx_1 + \sqrt{1 - t^2}\bar{z}$$

where $\bar{z} \sim \mathcal{N}(0, I_d)$ is another standard Gaussian variable independent of $x_1$. Diffusion models (Song et al., 2021) employ the Ornstein–Uhlenbeck (OU) SDE:

$$dY_s = -Y_s ds + \sqrt{2}dW_s, \quad Y_0 \sim p_{\text{data}}$$

which gradually adds noise to the data distribution $p_{\text{data}}$. Given $Y_0$, $Y_s$ can be written as

$$Y_s = e^{-s}Y_0 + \sqrt{1 - e^{-2s}}Z, \quad Z \sim \mathcal{N}(0, I_d).$$

Therefore, if we choose $x_1 \sim p_{\text{data}}$ in the stochastic interpolant, $Y_s$ and $x_{e^{-s}}$ have the same distribution.

**Previous Theoretical Results** For SDE-based generative models, Albergo et al. (2023) provide the following KL error bound when an estimator $\hat{b}_F(t, X_t^F)$ is used instead of the true drift term $b_F(t, X_t^F)$ in the SDE. This bound, which can also be derived using Girsanov's Theorem (Chen et al., 2023d; Oko et al., 2023), is given by:

$$
\begin{aligned}
\text{KL}(\rho(1)\|\hat{\rho}(1)) \leq \frac{1}{4\epsilon} \int_0^1 \int_{\mathbb{R}^d} &\|\hat{b}_F(t, x) \\
&- b_F(t, x)\|^2 \rho(t, x)dxdt.
\end{aligned}
$$

This inequality establishes an upper bound on the distribution estimation error in terms of the error in estimating the drift function $b_F(t, X_t^F)$. This estimation error is evaluated with respect to the true underlying density $\rho(t, x)$.

The primary limitation of the existing results is the assumption that the SDEs can be solved exactly (e.g., Chen et al. 2024b; Albergo et al. 2023). However, obtaining exact solutions is often challenging in practice, leading to the use of discrete-time samplers for estimating these solutions. Yet, when an SDE is discretized, discretization error causes distribution estimation error, which ultimately invalidates previous results. Moreover, the choice of discretizations can have a significant impact on the convergence of the dynamics, (see, e.g., Wibisono et al. 2016), leaving the design of optimal discretization methods largely open.

## 4. Main Results

In this section, we present a novel analysis for the discrete-time stochastic interpolant framework. Specifically, we focus on the following formulation: given a schedule $\{t_k\}_{k=0}^N \subseteq [0, 1]$ where $t_0 < t_1 < \cdots < t_{N-1} < t_N$, we define an estimated process using the following SDE:

$$d\hat{X}_t^F = \hat{b}_F(t_k, \hat{X}_{t_k}^F)dt + \sqrt{2\epsilon}dW_t, \quad t \in [t_k, t_{k+1}]. \quad (6)$$

In practice, we can express this as:

$$
\begin{aligned}
X_{t_{k+1}}^F =& X_{t_k}^F + (t_{k+1} - t_k)\hat{b}_F(t_k, \hat{X}_{t_k}^F) \\
&+ \sqrt{2\epsilon(t_{k+1} - t_k)}w_k, \qquad w_k \in \mathcal{N}(0, I_d)
\end{aligned}
\quad (7)
$$

for $k = 0, 1, \ldots, N - 1$. Here $\hat{b}_F(t, X_t^F)$ represents an estimator of the true drift term $b_F$. Equation (6), or equivalently (7), corresponds to the Euler-Maruyama discretization of the continuous-time SDE (Chen et al., 2023a;b). In this paper, we refer to (6) as the estimated SDE, while (3) is referred to as the true SDE.

Below, we present our analysis for the discrete-time stochastic interpolant. We begin by introducing the main assumptions, which will be crucial for bounding the error between the estimated distribution and the true target distribution.

**Assumption 4.1.** The joint measure $\nu$ defined in the stochastic interpolants (1) satisfies

$$\mathbb{E}_{(x_0,x_1)\sim\nu} \|x_0 - x_1\|^8 < \infty,$$

and the interpolation function is such that

$$\mathbb{E}_{(x_0,x_1)\sim\nu} \|\partial_t^2 I(t, x_0, x_1)\|^2 \leq M_2 < \infty, \quad \forall t \in [0, 1].$$

The first moment bound assumes that the initial and target distributions, $\rho_0$ and $\rho_1$, are not excessively far apart. In fact, the inequality (2) further implies that $\mathbb{E}\|\partial_t I(t, x_0, x_1)\|^8 < \infty$ for any $t \in [0, 1]$. The second part of the assumption ensures that the time derivative of the interpolation function does not exhibit significant variations. Assumption 4.1 is similar to previous assumptions on stochastic interpolants, with the exception of the first part, which utilizes the eighth moment instead of the fourth moment (see Albergo et al. 2023 or Appendix A).

**Assumption 4.2.** The estimator $\hat{b}_F$ satisfies

$$\sum_{k=0}^{N-1} (t_{k+1} - t_k)\mathbb{E}\|\hat{b}_F(t_k, x_{t_k}) - b_F(t_k, x_{t_k})\|^2 \leq \varepsilon_{b_F}^2,$$

where the expectations is taken over $x_{t_k} \sim \rho(t_k)$.

Assumption 4.2 assumes that we have a sufficiently accurate estimator for the drift term $b_F(t, x)$ at the discretized time points. This assumption is analogous to common assumptions employed in the theoretical analysis of diffusion models (Benton et al., 2024; Chen et al., 2023a).

Now we are ready to present our main theorem.

**Theorem 4.3.** *Suppose we start with an initial distribution $\hat{\rho}(t_0)$ and evolve the process according to the estimated SDE (6) with $\epsilon = O(1)$ until time $t = t_N$. Let $\hat{\rho}(t_N)$ be the distribution obtained at time $t = t_N$. Denote the ground truth marginal distributions by $\{\rho(t)\}_{t\in[0,1]}$, and define $\bar{\gamma}_k = \min_{t\in[t_k,t_{k+1}]} \gamma(t)$. We have under Assumptions 4.1 and 4.2 that:*

$$\text{KL}(\rho(t_N)\|\hat{\rho}(t_N)) \lesssim \text{KL}(\rho(t_0)\|\hat{\rho}(t_0)) + \epsilon^{-1}\varepsilon_{b_F}^2$$

$$+ \epsilon^{-1}\sum_{k=0}^{N-1}(t_{k+1} - t_k)^3$$

$$\cdot \left[M_2 + \bar{\gamma}_k^{-6}d^3 + \bar{\gamma}_k^{-2}d\sqrt{\mathbb{E}\|x_0 - x_1\|^8}\right]$$

$$+ \sum_{k=0}^{N-1}(t_{k+1} - t_k)^2\bar{\gamma}_k^{-2}d\left[\sqrt{\mathbb{E}\|x_0 - x_1\|^4} + \bar{\gamma}_k^{-2}d\right]$$

*Here the notation $f \lesssim g$ denotes $f = O(g)$.*

Theorem 4.3 provides the first finite-time error bound for the discrete-time stochastic interpolant framework (3), i.e., SDE (6). It explicitly quantifies the impact of the initial distribution mismatch (i.e., $\text{KL}(\rho(t_0)\|\hat{\rho}(t_0))$) and the $b_F$ estimation error (i.e., $\varepsilon_{b_F}^2$), demonstrates their dependence on the choice of latent scale $\gamma(t)$ and the time discretization schedule $\{t_k\}_{k=0}^N$. Notably, the bound offers a novel theoretical explanation for how the convergence behavior depends on the distance between the source and destination distributions, as reflected in the terms involving $\mathbb{E}\|x_0 - x_1\|^p$ with $p = 4$ and $p = 8$.

Now we explain the terms in Theorem 4.3.

The terms $(t_{k+1} - t_k)^3\left[M_2 + \bar{\gamma}_k^{-2}d\sqrt{\mathbb{E}\|x_0 - x_1\|^8}\right]$ and $(t_{k+1}-t_k)^2\bar{\gamma}_k^{-2}d\sqrt{\mathbb{E}\|x_0 - x_1\|^4}$ quantify the discretization error associated with the velocity function $v(t, x)$ (Equation (5)), which is a component of the drift term $b_F(t, x)$. Notably, the distance of form $\mathbb{E}\|x_0 - x_1\|^p$ is involved here, highlighting the influence of the distance between the source and target distributions on the discretization error.

Conversely, the terms $(t_{k+1} - t_k)^3\bar{\gamma}_k^{-6}d^3$ and $(t_{k+1} - t_k)^2\bar{\gamma}_k^{-4}d^2$ quantify the discretization error arising from the score function $s(t, x)$. This component of the discretization error exhibits a stronger dependence on the latent scale $\gamma(t)$ and the data dimension $d$. Specifically, assuming sufficiently small step sizes, the dependence of the score discretization error on $\gamma(t)$ is $\bar{\gamma}_k^{-4}$ (if we assume sufficiently small step sizes), which aligns with the findings for diffusion models (Chen et al., 2023a; Benton et al., 2024).

Since $\gamma(0) = \gamma(1) = 0$, the discretization error will become unbounded if we were to simulate the estimated SDE (6) from $t_0 = 0$ to $t_N = 1$. To address this, we choose $0 < t_0 < t_N < 1$ to ensure that $\bar{\gamma}_k$ is lower bounded, thereby maintaining a finite discretization error bound in Theorem 4.3. Under this approach, the SDE is simulated within the interval $[t_0, t_N]$, and an estimation of $\rho(t_N)$ is obtained instead of $\rho_1$. This is acceptable when $t_N$ is sufficiently close to 1, as $\rho(t_N)$ will be sufficiently close to $\rho_1$ (e.g., in terms of Wasserstein distance). This practice of choosing $t_N < 1$ is analogous to the early stopping technique commonly employed in diffusion models (Song et al., 2021; Chen et al., 2023a; Benton et al., 2024).

The term $\text{KL}(\rho(t_0)\|\hat{\rho}(t_0))$ quantifies the effect of choosing

a slightly different base distribution. As discussed earlier, we typically choose $t_0 > 0$, and in many cases, the true base distribution $\rho(t_0)$ may not be readily available. This bound theoretically supports the use of a similar base distribution $\hat{\rho}(t_0)$ as an approximation for the true base distribution.

Finally, the term $\varepsilon_{b_F}^2$ accounts for the error in estimating the drift term $b_F(t, x)$. Compared to previous continuous-time analyses of the stochastic interpolant framework, which typically measure the estimation error by an averaged error over the entire time interval (Albergo & Vanden-Eijnden, 2023; Albergo et al., 2023), our analysis evaluates the estimation error using a weighted average of the errors at the discretized time points.

The bound provided by Theorem 4.3 explicitly depends on the choice of latent scale $\gamma(t)$ and the time schedule $\{t_k\}_{k=0}^N$. This dependence can be leveraged to assess the computational complexity for a given time schedule under a specific choice of $\gamma(t)$. In Section 5, we will develop a time schedule that achieves a fast convergence rate.

We now provide a proof sketch for Theorem 4.3.

**Proof Sketch of Theorem 4.3** The proof of Theorem 4.3 contains two key steps. In step one, we establish a bound on the KL divergence due to discretization error in the drift term $b_F(t, x)$ of the SDE, based on Girsanov's theorem. Then, in step two, we exploit special structure of the $b_F(t, x)$ by expressing its derivatives as conditional covariances, enabling the application of relevant expectation inequalities and eventually bounding the discretization error.

**Step One: Bounding the KL-divergence with Discretization Error.** Leveraging the results provided by Chen et al. (2023d) (see Proposition C.2), which are derived using Girsanov's theorem, we obtain the following bound:

$$\text{KL}(\rho(t_N)\|\hat{\rho}(t_N))$$
$$\leq \text{KL}(\rho(t_0)\|\hat{\rho}(t_0)) + \text{KL}(P\|Q)$$
$$= \underbrace{\text{KL}(\rho(t_0)\|\hat{\rho}(t_0))}_{\text{Initialization error}}$$
$$+ \frac{1}{4\epsilon} \sum_{k=0}^{N-1} \int_{t_k}^{t_{k+1}} \mathbb{E}[\|b_F(t, X_t^F) - \hat{b}_F(t_k, X_{t_k}^F)\|^2] \mathrm{d}t$$

where $P$ and $Q$ represent the path measures of the solutions to the true SDE (3) and estimated SDE (6), respectively, both with the same initial distribution $\rho(t_0)$. Applying the triangle inequality yields:

$$\mathbb{E}[\|b_F(t, X_t^F) - \hat{b}_F(t_k, X_{t_k}^F)\|^2]$$
$$\leq 2 \underbrace{\mathbb{E}[\|b_F(t, X_t^F) - b_F(t_k, X_{t_k}^F)\|^2]}_{\text{Discretization error}}$$
$$+ 2 \underbrace{\mathbb{E}[\|b_F(t_k, X_{t_k}^F) - \hat{b}_F(t_k, X_{t_k}^F)\|^2]}_{\text{Estimation error}}$$

The second term on the right-hand-side corresponds to the estimation error of $\hat{b}_F(t, x)$, and its summation can be bounded by $\varepsilon_{b_F}^2$ according to Assumption 4.2. The first term, on the other hand, represents the discretization error associated with $b_F(t, x)$ and requires further analysis.

**Step Two: Bound the Discretization Error.** We now bound the discretization error above. A central tool in this part is the Itô's formula. By applying Itô's formula, we obtain:

$$\int_{t_k}^t \mathrm{d}b_F(s, X_s^F) = \int_{t_k}^t \partial_s b_F(s, X_s^F) \mathrm{d}s$$
$$+ \int_{t_k}^t \nabla b_F(s, X_s^F) \cdot b_F(s, X_s^F) \mathrm{d}s$$
$$+ \int_{t_k}^t \epsilon \Delta b_F(s, X_s^F) \mathrm{d}s \qquad (8)$$
$$+ \int_{t_k}^t \sqrt{2\epsilon} \nabla b_F(s, X_s^F) \cdot \mathrm{d}W_s.$$

While the application of Itô's formula is analogous to that of Benton et al. (2024), we refrain from eliminating the three linear terms due to their more complex forms in the context of stochastic interpolants. Instead, we apply Jensen's inequality on the integrals with respect to time and apply Itô's isometry (Le Gall 2016, Equation 5.8) on the integral with respect to Brownian motion, so that we can bound the term by the derivatives of $b_F(t, x)$ (i.e., terms like $\partial_t b_F(t, x)$ and $\nabla_x b_F(t, x)$, see Lemma C.1).

Since $b_F(t, x)$ can be expressed as a linear combination of $v(t, x)$ and $s(t, x)$, it remains to bound the derivatives of both $v(t, x)$ and $s(t, x)$, respectively. Note that $v(t, x)$ and $s(t, x)$ can be written as the conditional expectations of $\partial_t I$ and $\gamma^{-1} z = \frac{x_t - I}{\gamma^2}$ given $x_t = x$. To bound the derivatives of these conditional expectations, we employ the following key equality:

$$\partial_\alpha \mathbb{E}[f | x_t = x] = \mathbb{E}[\partial_\alpha f | x_t = x]$$
$$+ \text{Cov}(f, \partial_\alpha[-\|x - I\|^2 / 2\gamma^2] | x_t = x).$$

Here $\alpha$ can represent either $x$ or $t$. This equality crucially relies on the Gaussian latent term $\gamma(t)z$ introduced in the stochastic interpolant framework. This generalizes the result for $s(t, x) = \nabla \log \rho(t, x) = \mathbb{E}[\gamma^{-2}(I - x_t) | x_t = x]$ in diffusion models, as found in previous works (see, e.g., Bortoli 2022; Benton et al. 2024). We apply this equality extensively and derive bounds for both $s(t, x)$ and $v(t, x)$, where the function $v(t, x)$ does not appear or appears in a much simpler form (such as $x$) in the context of diffusion model theories. Subsequently, we apply a series of inequalities to ultimately bound the expectation over $x_t$. Detailed derivations and proofs can be found in B.

# 5. Schedule Design for Faster Convergence

In Theorem 4.3, we provide an upper bound on the KL divergence from the target distribution to the estimated distribution for a general class of SDE-based generative models. Since the bound depends on the choice of latent scale $\gamma(t)$ and schedule $\{t_k\}_{k=0}^N$, we are able to carefully design a time schedule for a given latent scale, thereby achieving a provably bounded error within a minimum number of steps.

Specifically, we consider the common choice of latent scale in stochastic interpolants, $\gamma(t) = \sqrt{at(1-t)}$, which is first introduced in Albergo et al. (2023). This choice is equivalent to changing the definition

$$x_t = I(t, x_0, x_1) + \gamma(t)z$$

to

$$x_t = I(t, x_0, x_1) + \sqrt{a}dB_t,$$

where $B_t$ is a standard Brownian bridge process independent of $(x_0, x_1)$. For this $\gamma(t)$, we present the following time schedule to optimize the sample complexity.

**Exponentially Decaying Time Schedule**  As suggested by Theorem 4.3, smaller steps need to be taken in order to balance the error terms. Moreover, to exactly cancel the $\gamma$-terms, we need $h_k = O(\bar{\gamma}_k^2)$ where $\bar{\gamma}$ is defined in Theorem 4.3. Hence, we propose an exponentially decaying time schedule inspired by the approach of Benton et al. (2024). Specifically, we first select a midpoint $t_M = \frac{1}{2}$. Let $h \in (0, 1)$ be a parameter that controls the step size. We then define the time steps as follows:

$$t_{k+1} - t_k = \begin{cases} \frac{1}{2}h(1-h)^{M-k-1}, & k < M \\ \frac{1}{2}h(1-h)^{k-M}, & k \geq M. \end{cases}$$

This leads to

$$t_k = \begin{cases} \frac{1}{2}(1-h)^{M-k}, & k < M \\ 1 - \frac{1}{2}(1-h)^{k-M}, & k \geq M. \end{cases}$$

The parameter $h$ determines the overall scale of the step sizes. A smaller $h$ results in a finer discretization of the time interval.

Let $h_k = t_{k+1} - t_k$ denote the step size at the $k$-th step. We observe that

$$h_k = O(h \min\{t_k, 1 - t_{k+1}\}) = O(h\bar{\gamma}_k^2),$$

which satisfies the condition of canceling the $\gamma$-terms. Moreover, the total number of steps is given by

$$N = O\left(\frac{\log(1/t_0) + \log(1/(1-t_N))}{\log(1/(1-h))}\right)$$
$$= O\left(h^{-1}\log\left(\frac{1}{t_0(1-t_N)}\right)\right).$$

Now we can provide the following bound:

**Proposition 5.1.** *Consider the same settings as in Theorem 4.3. Suppose $h_k = t_{t+1} - t_k = O(h\bar{\gamma}^2)$, $\epsilon = \Theta(1)$ and $h = O(\frac{1}{d})$. Then, we have*

$$\mathrm{KL}(\rho(t_N)\|\hat{\rho}(t_N)) \lesssim \epsilon^{-1}\varepsilon_{b_F}^2 + \mathrm{KL}(\rho(t_0)\|\hat{\rho}(t_0))$$
$$+ hd\sqrt{\mathbb{E}\|x_0 - x_1\|^4} + Nh^2d^2.$$

Proposition 5.1 provides the KL error bound when the step sizes is chosen so that the $\gamma$-terms are canceled.

**Corollary 5.2.** *Using $\gamma = \sqrt{at(1-t)}$ and the time schedule defined above, suppose that $\mathrm{KL}(\rho(t_0)\|\hat{\rho}(t_0)) \leq \varepsilon^2$ and $\varepsilon_{b_F}^2 \leq \epsilon \cdot \varepsilon^2$. Furthermore, assume that $\epsilon = \Theta(1)$ and $h = O(\frac{1}{d})$. Then, under the same settings as in Theorem 4.3, the total number of steps required to achieve $\mathrm{KL}(\rho(t_N)\|\hat{\rho}(t_N)) = O(\varepsilon^2)$ is:*

$$N = O\left\{\frac{1}{\varepsilon^2}\left[\sqrt{\mathbb{E}\|x_0 - x_1\|^4}d\log\left(\frac{1}{t_0(1-t_N)}\right)\right.\right.$$
$$\left.\left. + d^2\log^2\left(\frac{1}{t_0(1-t_N)}\right)\right]\right\}.$$

Corollary 5.2 provides the computational complexity of sampling data using the forward SDE. For a fixed error bound $\varepsilon$, the complexity scales proportionally to $\varepsilon^{-2}$. We can further decompose the complexity into distance-related complexity and Gaussian diffusion complexity. Here $O\left(\frac{1}{\varepsilon^2}\sqrt{\mathbb{E}\|x_0 - x_1\|^4}d\log\left(\frac{1}{t_0(1-t_N)}\right)\right)$ is the distance-related complexity representing the number of steps required to achieve a sufficiently small discretization error associated with the velocity function $v(t, x)$. $O\left(\frac{1}{\varepsilon^2}d^2\log^2\left(\frac{1}{t_0(1-t_N)}\right)\right)$ is the Gaussian diffusion complexity representing the number of steps required to achieve a sufficiently small discretization error associated with the score function $s(t, x)$.

We briefly explain how to obtain this complexity. First, given a desired number of steps $N$, we select

$$h = \Theta\left(N^{-1}\log\left(\frac{1}{t_0(1-t_N)}\right)\right)$$

to achieve the specified number of steps. Since $h_k = O(\bar{\gamma}_k^2 h)$, we have:

$$\sum_{k=0}^{N-1} h_k^3\left[M_2 + \bar{\gamma}_k^{-6}d^3 + \bar{\gamma}_k^{-2}d\sqrt{\mathbb{E}\|x_0 - x_1\|^8}\right]$$
$$\leq Nh^3d^3 + h^2\left(M_2 + d\sqrt{\mathbb{E}\|x_0 - x_1\|^8}\right),$$

and

$$\sum_{k=0}^{N-1}\left(h^2d^2 + h_khd\sqrt{\mathbb{E}\|x_0 - x_1\|^4}\right)$$
$$\leq Nh^2d^2 + hd\sqrt{\mathbb{E}\|x_0 - x_1\|^4}.$$

By substituting the chosen value of $h$ for the given $N$ into Theorem 4.3, we can derive the stated complexity bound.

**Comparison to a Uniform Schedule.** To highlight the benefits of our proposed exponentially decaying time schedule, we compare it with a natural uniform schedule that satisfies $h_k = \frac{t_N - t_0}{N} \approx \frac{1}{N}$. We further assume the ideal case where $\varepsilon_{b_F}^2 = 0$ and $\rho(t_0) = \hat{\rho}(t_0)$ in our analysis.

According to Theorem 4.3, the error bound for the uniform schedule is given by

$$\sum_{k=0}^{N-1} h_k^3(M_2 + \bar{\gamma}_k^{-6}d^3 + \bar{\gamma}_k^{-2}d\sqrt{\mathbb{E}\|x_0 - x_1\|^8})$$
$$+ \sum_{k=0}^{N-1} h_k^2(\bar{\gamma}_k^{-4}d^2 + \bar{\gamma}_k^{-2}d\sqrt{\mathbb{E}\|x_0 - x_1\|^4}).$$

Since $\bar{\gamma}_k^2 = \Theta(\min\{t_k, 1 - t_{k+1}\})$, and noting that

$$\int_\delta^{0.5} t^{-p}\mathrm{d}t = \begin{cases} \Theta(\log(1/\delta)), & p = 1 \\ \Theta(\delta^{-(p-1)}), & p > 1 \end{cases}$$

for a uniform schedule, the overall error bound becomes:

$$\mathrm{KL}(\rho(t_N)\|\hat{\rho}(t_N))$$
$$= O\left(\frac{1}{N}\left[\sqrt{\mathbb{E}\|x_0 - x_1\|^4}d\log\left(\frac{1}{t_0(1 - t_N)}\right)\right.\right.$$
$$\left.\left. + \frac{1}{t_0(1 - t_N)}d^2\right]\right).$$

Consequently, the complexity of using a uniform schedule is given by

$$N = O\left(\varepsilon^{-2}\left[\log\left(\frac{1}{t_0(1 - t_N)}\right)d\sqrt{\mathbb{E}\|x_0 - x_1\|^4}\right.\right.$$
$$\left.\left. + \frac{1}{t_0(1 - t_N)}d^2\right]\right),$$

which exhibits a higher computational complexity compared to the proposed exponentially decaying schedule.

**Comparison to Diffusion Models Results.** By setting $I(t, x_0, x_1) = (1 - t)x_0 + tx_1$, $\gamma(t) = \sqrt{2t(1 - t)}$, $x_0 \sim \rho_0 = \mathcal{N}(0, I_d)$, and assuming that $x_0$ and $x_1$ are independent, the stochastic interpolant reduces to $x_t = \sqrt{1 - t^2}\bar{z} + tx_1$ for some $\bar{z} \sim \mathcal{N}(0, I_d)$, which fits the diffusion model setting (Song et al., 2021). Assuming that the fourth moment of $\rho_1$ is bounded by a constant (see Appendix C.5 for details), the complexity of our approach simplifies to

$$N = O\left(\varepsilon^{-2}d^2\log^2\left(\frac{1}{1 - t_N}\right)\right).$$

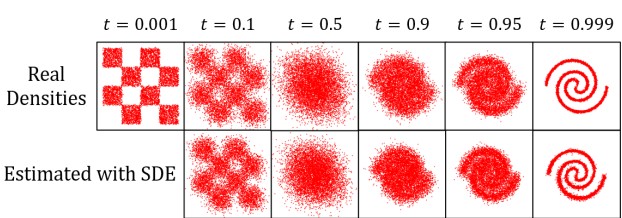

*Figure 1.* An illustration of the interpolants. We choose $I(t, x_0, x_1) = (1 - t)x_0 + tx_1$ and $\gamma(t) = \sqrt{2t(1 - t)}$. The first row of graphs shows the densities given by (1). The second row of graphs shows the estimated densities using SDE (6), where the estimator $\hat{b}_F$ is learned by a neural network.

For diffusion models with an early stopping time $\delta$, Chen et al. (2023a) established a complexity bound of $\tilde{O}\left(\varepsilon^{-2}d^2\log^2\left(\frac{1}{\delta}\right)\right)$. By setting $\delta = 1 - t_N$ in our analysis, we recover the same complexity bound as that obtained for diffusion models. While Benton et al. (2024) further improves the complexity bound for diffusion models to $\tilde{O}\left(\varepsilon^{-2}d\log^2\left(\frac{1}{\delta}\right)\right)$ by leveraging techniques from stochastic localization, these techniques heavily rely on the Gaussian structure of diffusion models and cannot be directly applied to the more general stochastic interpolant framework.

**Other Choices of $\gamma(t)$.** In addition to the commonly used $\gamma(t) = \sqrt{at(1 - t)}$, our framework can readily be extended to analyze other choices of $\gamma(t)$. In Appendix C.6, we present an analysis for $\gamma^2(t) = (1 - s)^2 s$, which is equivalent to the definition in Chen et al. (2024b). We show that the proposed time schedule in Appendix C.6 also outperforms the uniform schedule in terms of computational complexity, demonstrating the effectiveness of our schedule design.

## 6. Numerical Experiments

In this section, we conduct experiments to validate our theoretical results. We implement the discretized sampler as defined in Equation (7), and evaluate its performance on on two-dimensional datasets (primarily from Grathwohl et al. 2019) and Gaussian mixtures. The experiments focus on the following three aspects: (i) the convergence rate using the exponentially decaying schedule, (ii) the effect of choosing different $\rho_0$ as base densities, which is reflected by the distribution distance terms in our bound, and (iii) the dependence of the error bound on $d$.

We employ $I(t, x_0, x_1) = tx_1 + (1 - t)x_0$, $\gamma(t) = \sqrt{2t(1 - t)}$ and $\epsilon = 1$ in our experiments. Figure 1 presents a visualization of the interpolants and the estimated densities generated using the forward SDE. As we can see in Figure 1, the density defined by the stochastic interpolant progressively changes from the checkerboard density to the spiral density, and the estimated density given by SDE (6)

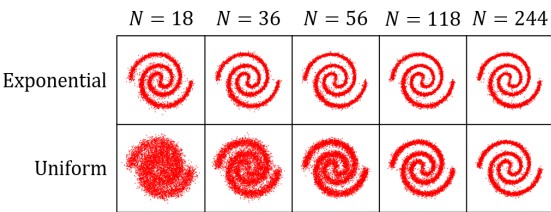

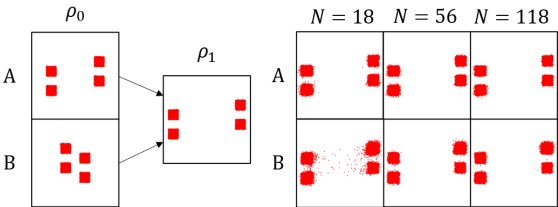

(a) An illustration of generation results using different schedules.

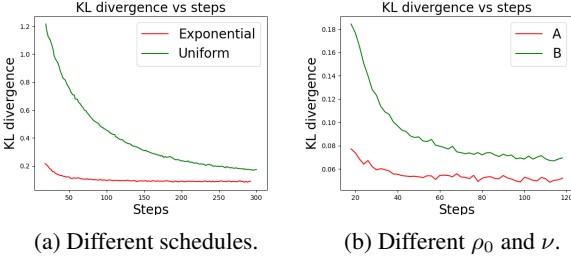

(b) A view of different $\rho_0$ and their effects on the convergence rate.

*Figure 2.* A visualization of experiments.

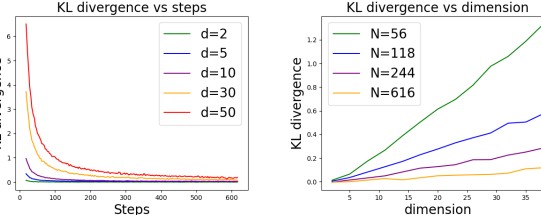

(a) Different schedules.   (b) Different $\rho_0$ and $\nu$.

*Figure 3.* Estimated KL divergences. The $x$-axis refers to the number of steps, while the $y$-axis refers to the empirical KL divergence between target distribution $\rho(t_N)$ and estimated distribution $\hat{\rho}(t_N)$. Figure 3a compares the exponentially decaying schedule (red line) with the uniform schedule (green line). Figure 3b compares different settings of $\rho_0$ and $\nu$ (setting A: red line; setting B: green line).

well tracks the change of the real density.

**Comparison of Different Time Schedules** We now compare the performance of different schedules, namely, those proposed in Section 5 and the uniform step sizes. The task involves transforming from a "checkerboard" density to a "spiral" density. As illustrated in Figure 2a, employing exponentially decaying step sizes results in about $10\times$ faster convergence of the target density compared to using uniform step sizes.

To quantitatively assess this, we estimate the KL divergence between the target density and the generated densities using sampled data points, and we put the details of the estimation of KL divergence in Appendix D. The lower bound observed in the estimated distances is attributed to the estimation error $\varepsilon_{b_F}^2$ and the inherent randomness in the TV distance estimation process. Both Figure 2a and Figure 3a corrobo-

rate the superior convergence performance of exponentially decaying step sizes.

**Effect of Different Distribution Distances** Second, we investigate the impact of different source densities ($\rho_0$) and couplings ($\nu$) on the complexity of the generation process while keeping the target density ($\rho_1$) fixed. The densities are shown in 2b, where we have two source densities, namely "A" and "B". Each "block" in $\rho_0$ is coupled with the corresponding "block" in $\rho_1$ in the same vertical position.

Figure 3b shows the estimated TV distances between the generated data distribution and the target distribution for both scenarios. For both choices of $\rho_0$, we utilized the exponentially decaying step size schedule. The results demonstrate that the generation process converges faster when the source density ($\rho_0$) is closer to the target density ($\rho_1$) under the specified coupling ($\nu$). In contrast, the generation process converges slower when the source density is farther away from the target density. This observation highlights the influence of the initial distribution and the coupling structure on the overall convergence behavior.

**Test on $d$-dimensional Gaussian Mixtures** Next, we compare the performance of the sampler under different dimension $d$. The source and target distribution are chosen as Gaussian mixtures, as in this case the drift $b_F(t, x)$ has an analytical form (Albergo et al., 2023). The comparison is shown in Figure 4.

(a) Convergence for different $d$.   (b) Error for fixed $N$.

*Figure 4.* The impact of dimension $d$. Figure 4a shows the convergence for different $d$. Figure 4b compares the KL error for different $d$ under fixed number of iterations $N$.

The upper bound of the KL error provided by our theory is $O(d^2)$, but the empirical KL error evaluated in this experiment grows almost linearly w.r.t. $d$. This gap could be due to several reasons. For example, there might be some special properties of Gaussian mixtures which do not hold for general distributions, or there simply exists a sharper bound that we have not found. Filling this gap would be an interesting direction for future research.

## 7. Conclusion and Future Directions

This paper provides the first discrete-time analysis for the SDE-based generative models within the stochastic inter-

polant framework. We formulate a discrete-time sampler using the Euler–Maruyama scheme to estimate the target distribution by leveraging learned velocity estimators. We then provide an upper bound on the KL divergence from the target distribution to the estimated distribution. Our result provides a novel quantification on how different factors, including the distance between source and target distributions $\sqrt{\mathbb{E}_\nu \|x_0 - x_1\|^p}$ and the desired estimation accuracy $\varepsilon^2$, affect the convergence rate and also offers a new principled way to design efficient schedule for convergence acceleration. Finally, we also conduct numerical experiments with the discrete-time sampler, which validates our theoretical findings.

Future research avenues can fruitfully explore the enhancement of convergence bounds, with a particular focus on addressing the dependency on the dimension $d$. Notably, diffusion models are demonstrated by previous works to exhibit near $d$-linear convergence rates, indicating the potential for improvement. Another direction is to investigate strategies for refining the sampling algorithm to attain convergence rates superior to the currently observed $O(\varepsilon^{-2})$. Furthermore, the finite-time convergence phenomenon of ODE-based generative models within the context of the stochastic interpolant framework warrants a more comprehensive investigation.

## Grant Acknowledgement

This work is supported by the National Natural Science Foundation of China Grants 52450016 and 52494974.

## Impact Statement

This paper presents work whose goal is to advance the field of Machine Learning. There are many potential societal consequences of our work, none which we feel must be specifically highlighted here.

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

# Appendices

- **Appendix A: Supplementary details for Section 3** - This section provides additional details from (Albergo et al., 2023) that were not included in Section 3. It includes a formal statement of key equations and their associated conditions. Furthermore, it outlines the optimization objectives for the score and velocity estimators, which are used for model training in Section 6.

- **Appendix B: Useful lemmas in bounding derivatives** - This section presents lemmas that are essential for bounding the derivatives of velocity functions and score functions, along with their corresponding proofs. The proofs primarily rely on properties and inequalities related to (conditional) expectations. These lemmas play a crucial role in deriving the overall KL error bounds.

- **Appendix C: Proofs of results in Section 4 and 5** - This section provides the complete proofs for the results presented in Section 4 and Section 5. This includes the proof of Theorem 4.3 (Appendix C.2), the proof of Proposition 5.1 and Corollary 5.2 (Appendices C.3,C.4), and additional details regarding the discussions in Section 5 (Appendix C.6).

- **Appendix D: More details of numerical experiments** - This section provides omitted details for Section 6, including the parameterization of estimators, choice of $(t_0, t_N)$ and optimizers, and how the TV distance is estimated. We also include additional experiments for $\gamma(t) = \sqrt{(1-t)^2 t}$.

## Notations

We use $\| \cdot \|$ to denote $\ell_2$ norm for both vectors and matrices. For a matrix $A$, we use $\|A\|_F = \sqrt{\sum_{ij} A_{ij}^2}$ to denote the Frobenious norm of $A$. We use $\frac{\mathrm{d}}{\mathrm{d}u}$, $\frac{\partial}{\partial u}$, or just $\partial_u$ to denote the (partial) derivative with respect to $u$. We use $\nabla$ to denote the gradient or Jacobian, depending on whether the function is scalar-valued or vector-valued. If not specified, for the function in form of $f(t, x)$ where $t$ is a scalar and $x$ is a vector, $\nabla f(t, x)$ means the gradient vector or Jacobian matrix with respect to $x$ rather than $t$. We use $\Delta f(t, x) = \sum_{i=1}^d \frac{\partial^2}{\partial x_i^2} f$ as the Laplace operator. We use $\mathbb{E}[X]$ to denote the expectation of a random variable $X$, and $\mathrm{Cov}(X, Y)$ to denote the covariance of two random variables $X, Y$. $\mathbb{E}[X|c]$ and $\mathrm{Cov}(X, Y|c)$ denote the corresponding conditional expectation and conditional covariance given condition $c$. We use the notation $f(x) \lesssim g(x)$ or $f(x) = O(g(x))$ to denote that there exists a constant $C > 0$ such that $f(x) \leq Cg(x)$.

## A. Supplementary Details for Section 3

This part summarizes some of the results from (Albergo et al., 2023) that are not introduced in Section 3.

**Proposition A.1.** *((Albergo et al., 2023), Theorem 2.6, Corollaries 2.10 and 2.18, and their proofs) Suppose that the joint measure $\nu$ and the function $I$ satisfies*

$$\underset{(x_0, x_1) \sim \nu}{\mathbb{E}} \|\partial_t I(t, x_0, x_1)\|^4 \leq M_1 < \infty, \qquad \underset{(x_0, x_1) \sim \nu}{\mathbb{E}} \|\partial_t^2 I(t, x_0, x_1)\|^2 \leq M_2 < \infty, \quad \forall t \in [0, 1]. \tag{9}$$

*Then, $\rho \in C^1((0, 1), C^p(\mathbb{R}^d))$, $s \in C^1((0, 1), (C^p(\mathbb{R}^d))^d)$ and $b \in C^0((0, 1), (C^p(\mathbb{R}^d))^d)$, and both the solution of the probability flow ODE*

$$\frac{\mathrm{d}}{\mathrm{d}t} X_t = b(t, X_t), \qquad X_0 \sim \rho_0$$

*and the solution of the forward SDE*

$$\mathrm{d}X_t^F = b_F(t, X_t^F)\mathrm{d}t + \sqrt{2\epsilon(t)}\mathrm{d}W_t, \qquad X_0^F \sim \rho_0$$

*have the same marginal densities as $(x_t)_{t \in [0,1]}$. Here $\epsilon \in C[0, 1]$ with $\epsilon(t) \geq 0$ for all $t \in [0, 1]$ and $b_F$ is defined as*

$$b_F(t, x) = b(t, x) + \epsilon(t)s(t, x). \tag{10}$$

*Moreover, suppose that the densities $\rho_0, \rho_1$ are strictly positive elements of $C^2(\mathbb{R}^d)$, and are such that*

$$\int_{\mathbb{R}^d} \|\nabla \log \rho_0(x)\|^2 \rho_0(x) dx < \infty, \qquad \int_{\mathbb{R}^d} \|\nabla \log \rho_0(x)\|^2 \rho_1(x) dx < \infty.$$

*Then $\rho \in C^1([0, 1], C^p(\mathbb{R}^d))$, $s \in C^1([0, 1], (C^p(\mathbb{R}^d))^d)$ and $b \in C^0([0, 1], (C^p(\mathbb{R}^d))^d)$. The notation is adapted from (Albergo et al., 2023) where $f \in C^1([0, 1], C^p(\mathbb{R}^d))$ means that the function $f$ is $C^1$ in $t \in [0, 1]$ and $C^p$ in $x \in \mathbb{R}^d$.*

The above proposition provides a generative modeling in the form of

$$\frac{\mathrm{d}}{\mathrm{d}t} X_t = b(t, X_t)$$

and

$$\mathrm{d}X_t^F = b_F(t, X_t^F)\mathrm{d}t + \sqrt{2\epsilon(t)}\mathrm{d}W_t.$$

In practice, we need to train an estimator to estimate velocity functions. By the following proposition, we can use the optimization objectives to train the estimators.

**Proposition A.2.** *((Albergo et al., 2023), Theorems 2.7 and 2.8) $b$ is the unique minimizer of*

$$\mathcal{L}_b[\hat{b}] = \int_0^1 \mathbb{E}\left[\frac{1}{2}\|\hat{b}(t, x_t)\|^2 - (\partial_t I(t, x_0, x_1) + \dot{\gamma}(t)z) \cdot \hat{b}(t, x_t)\right] \mathrm{d}t,$$

*and $s$ is the unique minimizer of*

$$\mathcal{L}_s[\hat{s}] = \int_0^1 \mathbb{E}\left[\frac{1}{2}\|\hat{s}(t, x_t)\|^2 + \gamma^{-1}(t)z \cdot \hat{s}(t, x_t)\right] \mathrm{d}t.$$

*Here the notation "$\cdot$" represents the inner product of two vectors.*

## B. Bounding the Velocities and Scores

### B.1. Useful Lemmas

To begin with, we first provide moment bounds on the Gaussian variable $z \sim \mathcal{N}(0, I_d)$.

**Lemma B.1.** *For any $p \geq 1$,*

$$\mathbb{E}\|z\|^{2p} \leq C(p)d^p,$$

*where $C(p)$ is a constant that only depends on $p$.*

*Proof.* First, $\|z\|^2 = \sum_{i=1}^n z_i^2$, where we represent $z = (z_1, z_2, \cdots, z_d)^T$. For any $n$ positive numbers $a_1, a_2, \ldots, a_n$, using Jensen's inequality,

$$\left(\sum_{i=1}^n a_i\right)^p = n^p \left(\frac{1}{n}\sum_{i=1}^n a_i\right)^p \leq n^p \cdot \frac{1}{n}\sum_{i=1}^n a_i^p.$$

Then,

$$
\begin{aligned}
\mathbb{E}\|z\|^{2p} &= \mathbb{E}\left[\left(\sum_{i=1}^d z_i^2\right)^p\right] \\
&\leq d^p \cdot \frac{1}{d}\sum_{i=1}^d \mathbb{E}[|z_i|^{2p}] \quad \text{(Jensen's inequality)} \\
&\leq d^p \mathbb{E}\left[|z_1|^{2p}\right] \quad\quad (\{z_i\}_{i=1}^d \text{ are i.i.d.}) \\
&= C(p)d^p.
\end{aligned}
$$

Here the constant

$$C(p) = \int_{-\infty}^{\infty} \frac{1}{\sqrt{2\pi}} e^{-\frac{x^2}{2}} |x|^{2p} \mathrm{d}x < \infty$$

only depends on $p$. $\qquad\square$

Also, the following is another simple fact that is useful for our analysis.

**Lemma B.2.** *For two vectors $u \in \mathbb{R}^n$, $v \in \mathbb{R}^m$, the matrix $uv^T \in \mathbb{R}^{n \times m}$ satisfies*

$$\|uv^T\|_F = \|u\| \cdot \|v\|,$$

*where $\|\cdot\|_F$ denotes the Frobenious norm and $\|\cdot\|$ denotes the 2-norm.*

*Proof.* By the definition of the Frobenious norm,

$$
\begin{aligned}
\|uv^T\|_F^2 &= \sum_{i=1}^n \sum_{j=1}^m (uv^T)_{ij}^2 \\
&= \sum_{i=1}^n \sum_{j=1}^m u_i^2 v_j^2 \\
&= \sum_{i=1}^n u_i^2 \cdot \sum_{j=1}^m v_j^2 \\
&= \|u\|^2 \cdot \|v\|^2.
\end{aligned}
$$

$\qquad\square$

Recall that we have defined $v(t, x) = \mathbb{E}[\partial_t I(t, x_0, x_1)|x_t = x]$. We then give bounds for the score functions and the velocity functions.

**Lemma B.3.** *For $p \geq 1$, there exists a constant $C(p)$ that depends only on $p$, s.t. for $t \in (0,1)$,*

$$\mathbb{E}\|s(t, x_t)\|^p \leq C(p)\gamma^{-p}d^{p/2},$$
$$\mathbb{E}\|v(t, x_t)\|^p \leq C(p)\mathbb{E}\|x_1 - x_0\|^p,$$
$$\mathbb{E}\|b(t, x_t)\|^p \leq C(p)\left[\mathbb{E}\|x_1 - x_0\|^p + \dot{\gamma}d^{p/2}\right],$$
$$\mathbb{E}\|b_F(t, x_t)\|^p \leq C(p)\left[\mathbb{E}\|x_1 - x_0\|^p + (\dot{\gamma}^p - \gamma^{-p}\epsilon^p)d^{p/2}\right].$$

*Proof.* When $p \geq 1$, use the conditional expectation form of $s$ and $v$ and apply Jensen's inequality, we then obtain

$$\mathbb{E}\|s(t, x_t)\|^p = \mathbb{E}\|\gamma^{-1}\mathbb{E}[z|x_t = x]\|^p \leq \gamma^{-p}\mathbb{E}\|z\|^p \leq C(p)\gamma^{-p}d^{p/2},$$
$$\mathbb{E}\|v(t, x_t)\|^p = \mathbb{E}\|\mathbb{E}[\partial_t I|x_t = x]\|^p \leq \mathbb{E}\|\partial_t I\|^p \leq C(p)\mathbb{E}\|x_1 - x_0\|^p,$$

Moreover, since $b(t, x) = v(t, x) + \dot{\gamma}\gamma s(t, x)$ and $b_F(t, x) = b(t, x) + \epsilon s(t, x)$,

$$\mathbb{E}\|b(t, x_t)\|^p \leq C(p)\left[\mathbb{E}\|x_1 - x_0\|^p + \dot{\gamma}^p d^{p/2}\right],$$
$$\mathbb{E}\|b_F(t, x_t)\|^p \leq C(p)\left[\mathbb{E}\|x_1 - x_0\|^p + (\dot{\gamma}^p - \gamma^{-p}\epsilon^p)d^{p/2}\right].$$

$\square$

## B.2. Bounds on Time and Space Derivatives

**Note:** In the following sections, we will use the fact that $\frac{d}{dt}\gamma^2(t) = O(1)$ and $\frac{d^2}{dt^2}\gamma^2(t) = O(1)$.

Before we move on to the lemmas, we first discuss the conditional expectation itself. By the definition $x_t = I(t, x_0, x_1) + \gamma(t)z$, we can just know that the density of $x_t$ can be expressed as

$$\rho(t, x) = \int_{\mathbb{R}^d \times \mathbb{R}^d} \frac{1}{(2\pi\gamma(t)^2)^{d/2}} \exp\left(-\frac{\|x - I(t, x_0, x_1)\|^2}{2\gamma(t)^2}\right) d\nu(x_0, x_1).$$

Also, under the condition $x_t = x$, the conditional measure of $(x_0, x_1)$ is then

$$\frac{1}{\rho(t, x)} \cdot \frac{1}{(2\pi\gamma(t)^2)^{d/2}} \exp\left(-\frac{\|x - I(t, x_0, x_1)\|^2}{2\gamma(t)^2}\right) d\nu(x_0, x_1).$$

Therefore, for any function $f_t(x_t, x_0, x_1)$, its conditional expectation can be written as

$$\mathbb{E}[f_t(x_t, x_0, x_1)|x_t = x] = \int_{\mathbb{R}^d \times \mathbb{R}^d} \frac{f_t(x, x_0, x_1)}{\rho(t, x)} \cdot \frac{1}{(2\pi\gamma(t)^2)^{d/2}} \exp\left(-\frac{\|x - I(t, x_0, x_1)\|^2}{2\gamma(t)^2}\right) d\nu(x_0, x_1)$$
$$= \frac{\mathbb{E}_{(x_0,x_1)\sim\nu}\left[\exp\left(-\frac{\|x - I(t, x_0, x_1)\|^2}{2\gamma(t)^2}\right) \cdot f_t(x, x_0, x_1)\right]}{\mathbb{E}_{(x_0,x_1)\sim\nu}\left[\exp\left(-\frac{\|x - I(t, x_0, x_1)\|^2}{2\gamma(t)^2}\right)\right]}.$$

We first consider the time derivative of $v$ in the sense of expectation.

**Lemma B.4.** *We have*
$$\mathbb{E}\|\partial_t v(t, x_t)\|^2 \lesssim \mathbb{E}\|\partial_t^2 I\|^2 + \gamma^{-2}d\mathbb{E}\|x_0 - x_1\|^4 + \gamma^{-2}\dot{\gamma}^4 d^3$$

*for $t \in (0, 1)$.*

*Proof.* For $t \in (0, 1)$, we can first explicitly write

$$v(t, x) = \frac{\mathbb{E}_{(x_0,x_1)\sim\nu}\left[\exp\left(-\frac{\|x - I(t)\|^2}{2\gamma(t)^2}\right) \cdot \partial_t I(t)\right]}{\mathbb{E}_{(x_0,x_1)\sim\nu}\left[\exp\left(-\frac{\|x - I(t)\|^2}{2\gamma(t)^2}\right)\right]}.$$

Here we write $I(t) = I(t, x_0, x_1)$ for simplicity, and below we will omit $t$ when it is clear in the context. We now want to compute $\partial_t v(t, x)$. First notice that

$$\frac{\mathrm{d}}{\mathrm{d}t}\left[\exp\left(-\frac{\|x - I\|^2}{2\gamma^2}\right) \cdot \partial_t I\right] = \exp\left(-\frac{\|x - I\|^2}{2\gamma^2}\right) \cdot \left[\partial_t^2 I + \partial_t I \cdot \left(\frac{\|x - I\|^2}{\gamma(t)^3}\dot\gamma + \frac{x - I}{\gamma^2} \cdot \partial_t I\right)\right].$$

Note that $\sup_{x \in \mathbb{R}} \exp(-x^2/2)x = e^{-1/2} = C_1 < \infty$, $\sup_{x \in \mathbb{R}} \exp(-x^2/2)x^2 = 2e^{-1} = C_2 < \infty$, we know that

$$\left\|\frac{\mathrm{d}}{\mathrm{d}t}\left[\exp\left(-\frac{\|x - I\|^2}{2\gamma^2}\right) \cdot \partial_t I\right]\right\| \leq \|\partial_t^2 I\| + C_2\gamma^{-1}\dot\gamma\|\partial_t I\| + C_1\gamma^{-1}\|\partial_t I\|^2,$$

Therefore, using dominated convergence theorem, we know that

$$\frac{\mathrm{d}}{\mathrm{d}t}\mathop{\mathbb{E}}_{(x_0,x_1)\sim\nu}\left[\exp\left(-\frac{\|x - I\|^2}{2\gamma^2}\right) \cdot \partial_t I\right] = \mathop{\mathbb{E}}_{(x_0,x_1)\sim\nu}\left[\frac{\mathrm{d}}{\mathrm{d}t}\left(\exp\left(-\frac{\|x - I\|^2}{2\gamma^2}\right) \cdot \partial_t I\right)\right].$$

Similarly we can do this for the denominator, so that we can compute the overall derivative. Let $f_t(x_0, x_1) = -\frac{\|x - I(t)\|^2}{2\gamma^2}$, for simplicity we may just write $f_t$. Then,

$$\begin{aligned}
\partial_t v(t, x) &= \frac{\mathop{\mathbb{E}}_{(x_0,x_1)\sim\nu}\left[\exp(f_t) \cdot \partial_t^2 I\right]}{\mathop{\mathbb{E}}_{(x_0,x_1)\sim\nu}\left[\exp(f_t)\right]} \\
&+ \frac{\mathop{\mathbb{E}}_{(x_0,x_1)\sim\nu}\left[\exp(f_t) \cdot \partial_t I \cdot \partial_t f_t\right]}{\mathop{\mathbb{E}}_{(x_0,x_1)\sim\nu}\left[\exp(f_t)\right]} \\
&- \frac{\mathop{\mathbb{E}}_{(x_0,x_1)\sim\nu}\left[\exp(f_t) \cdot \partial_t I\right] \cdot \mathop{\mathbb{E}}_{(x_0,x_1)\sim\nu}\left[\exp(f_t) \cdot \partial_t f_t\right]}{\left[\mathop{\mathbb{E}}_{(x_0,x_1)\sim\nu}\left[\exp(f_t)\right]\right]^2} \\
&= \mathbb{E}[\partial_t^2 I | x_t = x] \\
&+ \mathrm{Cov}(\partial_t I, \partial_t f_t | x_t = x),
\end{aligned}$$

where the last equality uses the previous explanations of conditional expectations. Hence,

$$\|\partial_t v(t, x)\| \leq \mathbb{E}[\|\partial_t^2 I\| | x_t = x] + \sqrt{\mathbb{E}[|\partial_t f_t|^2 | x_t = x]}\sqrt{\mathbb{E}[\|\partial_t I\|^2 | x_t = x]}.$$

Therefore, we have

$$\begin{aligned}
\mathbb{E}\|\partial_t v(t, x_t)\|^2 &\leq 2\mathbb{E}[\mathbb{E}[\|\partial_t^2 I\|^2 | x_t]] + 2\mathbb{E}[\mathbb{E}[|\partial_t f_t|^2 | x_t] \cdot \mathbb{E}[\|\partial_t I\|^2 | x_t]] && ((a+b)^2 \leq 2a^2 + 2b^2) \\
&\leq 2\mathbb{E}\|\partial_t^2 I\|^2 + 2\sqrt{\mathbb{E}[\mathbb{E}[|\partial_t f_t|^2 | x_t]^2]} \cdot \sqrt{\mathbb{E}[\mathbb{E}[\|\partial_t I\|^2 | x_t]^2]} && \text{(Cauchy-Schwarz inequality)} \\
&\leq 2\mathbb{E}\|\partial_t^2 I\|^2 + 2\sqrt{\mathbb{E}[\mathbb{E}[|\partial_t f_t|^4 | x_t]]} \cdot \sqrt{\mathbb{E}[\mathbb{E}[\|\partial_t I\|^4 | x_t]]} && \text{(Jensen's inequality)} \\
&\leq 2\mathbb{E}\|\partial_t^2 I\|^2 + 2\sqrt{\mathbb{E}|\partial_t f_t|^4}\sqrt{\mathbb{E}\|\partial_t I\|^4}.
\end{aligned}$$

Using the requirement $\partial_t I \leq C\|x_0 - x_1\|$ in the definition of stochastic interpolants, $\|\partial_t I\|^4 \lesssim \|x_0 - x_1\|^4$. For $\partial_t f_t$, we can directly obtain

$$\partial_t f = \frac{\|x - I\|^2}{\gamma^3}\dot\gamma + \gamma^{-2}(x - I) \cdot \partial_t I = \gamma^{-1}\dot\gamma\|z\|^2 + \gamma^{-1}z \cdot \partial_t I.$$

Recall that we have defined $x_t = I(t, x_0, x_1) + \gamma(t)z$ where $z$ is an independent gaussian variable $z \sim \mathcal{N}(0, I_d)$. By Lemma B.1,

$$\mathbb{E}\|z\|^8 \lesssim d^4, \qquad \mathbb{E}\|z\|^4 \lesssim d^2,$$

we have

$$\mathbb{E}|\partial_t f_t|^4 \lesssim (\gamma^{-1}\dot\gamma)^4 d^4 + \gamma^{-4}d^2\mathbb{E}\|x_0 - x_1\|^4.$$

Therefore, we can finally deduce that

$$\mathbb{E}\|\partial_t v(t, x_t)\|^2 \lesssim \mathbb{E}\|\partial_t^2 I\|^2 + \gamma^{-2} d\mathbb{E}\|x_0 - x_1\|^4 + \gamma^{-2}\dot{\gamma}^4 d^3.$$

$\square$

In addition, we want to consider the space derivative of the velocity for a fixed $t \in (0, 1)$. That is, we want to give a bound for $\nabla v(t, x)$. Here we use the notation $\nabla v(t, x)$ to denote the Jacobian matrix $\left(\frac{d}{dx^i} v(t, x)_j\right)_{ij}$, where $x^i$ represents the value of vector $x$ at the $i$-th dimension.

**Lemma B.5.** *We have*

$$\mathbb{E}\|\nabla v(t, x)\|_F^p \leq C(p)\gamma^{-p} d^{p/2}\sqrt{\mathbb{E}\|x_0 - x_1\|^{2p}}$$

*for $p \geq 1$, $t \in (0, 1)$, where $C(p)$ is a constant that only depends on $p$ and $\|\cdot\|_F$ denotes the Frobenius norm.*

*Proof.* Similar to the proof of Lemma B.4,

$$\nabla\left(\exp\left(-\frac{\|x - I\|^2}{2\gamma^2}\right) \cdot \partial_t I\right) = \exp\left(-\frac{\|x - I\|^2}{2\gamma^2}\right)\left(\partial_t I \otimes \nabla\left(-\frac{\|x - I\|^2}{2\gamma^2}\right)\right),$$

where $\otimes$ denotes the tensor product, which denotes $\partial_t I \otimes \nabla\left(-\frac{\|x-I\|^2}{2\gamma^2}\right) = \partial_t I \cdot \nabla\left(-\frac{\|x-I\|^2}{2\gamma^2}\right)^T$ here in the matrix form. Again, by dominated convergence theorem we can move the gradient operator into the expectation. Using the same notations (i.e., $f_t$ and so on), we can deduce that

$$\nabla v(t, x) = \frac{\underset{(x_0, x_1) \sim \nu}{\mathbb{E}}[\exp(f_t) \cdot (\partial_t I \otimes \nabla f_t)]}{\underset{(x_0, x_1) \sim \nu}{\mathbb{E}}[\exp(f_t)]}$$

$$- \frac{\underset{(x_0, x_1) \sim \nu}{\mathbb{E}}[\exp(f_t) \cdot \partial_t I] \otimes \underset{(x_0, x_1) \sim \nu}{\mathbb{E}}[\exp(f_t) \cdot \nabla f_t]}{\left[\underset{(x_0, x_1) \sim \nu}{\mathbb{E}}[\exp(f_t)]\right]^2}$$

$$= \mathrm{Cov}(\partial_t I, \nabla f_t | x_t = x).$$

Again, the last equality uses the definition of covariance. Thus, by Cauchy-Schwarz inequality,

$$\|\nabla v(t, x)\|_F \leq \sqrt{\mathbb{E}[\|\partial_t I\|^2 | x_t = x]}\sqrt{\mathbb{E}[\|\nabla f_t\|^2 | x_t = x]}.$$

Therefore, we can use Cauchy-Schwarz inequality again and apply Jensen's inequality to deduce that for any $p \geq 1$,

$$\mathbb{E}\|\nabla v(t, x_t)\|_F^p \leq \sqrt{[\mathbb{E}[\mathbb{E}\|\partial_t I\|^2 | x_t]]^p} \cdot \sqrt{[\mathbb{E}[\mathbb{E}\|\nabla f_t\|^2 | x_t]]^p}$$

$$\leq \sqrt{\mathbb{E}\|\partial_t I\|^{2p}} \cdot \sqrt{\mathbb{E}\|\nabla f_t\|^{2p}}.$$

It is clear that $\mathbb{E}\|\partial_t I\|^{2p} \lesssim \mathbb{E}\|x_0 - x_1\|^{2p}$. Note

$$\nabla f_t = -\frac{x - I}{\gamma^2} = -\gamma^{-1} z,$$

we then deduce that

$$\mathbb{E}\|\nabla f_t\|^{2p} \leq C(p)\gamma^{-2p} d^p$$

for some constant that only depends on $p$. The lemma is then obtained. $\square$

Despite the function $v(t, x)$, we are also interested in the score function $s(t, x)$. The following lemmas provide some similar bounds for $s(t, x)$.

**Lemma B.6.**
$$\mathbb{E}\|\partial_t\left(\gamma s(t, x_t)\right)\|^2 \lesssim \gamma^{-2}\dot{\gamma}^2 d^3 + \gamma^{-2}d^2\sqrt{\mathbb{E}\|x_0 - x_1\|^4}$$

*and*
$$\mathbb{E}\|\partial_t s(t, x_t)\|^2 \lesssim \gamma^{-4}\dot{\gamma}^2 d^3 + \gamma^{-4}d^2\sqrt{\mathbb{E}\|x_0 - x_1\|^4}.$$

*for any $t \in (0, 1)$,*

*Proof.* First using the analysis for the conditional expectations, we obtain that

$$s(t, x) = \nabla \log \rho(t, x) = -\frac{\underset{(x_0, x_1)\sim\nu}{\mathbb{E}}\left[\exp\left(-\frac{\|x - I\|^2}{2\gamma^2}\right) \cdot \frac{x - I}{\gamma^2}\right]}{\underset{(x_0, x_1)\sim\nu}{\mathbb{E}}\left[\exp\left(-\frac{\|x - I\|^2}{2\gamma^2}\right)\right]}.$$

In order to compute $\partial_t(\gamma s(t, x))$, we apply a similar analysis as the proof of Lemma B.4 with exactly the same notations to deduce that

$$\partial_t s(t, x) = \frac{\underset{(x_0, x_1)\sim\nu}{\mathbb{E}}\left[\exp(f_t) \cdot \partial_t(\gamma\nabla f_t)\right]}{\underset{(x_0, x_1)\sim\nu}{\mathbb{E}}\left[\exp(f_t)\right]}$$

$$+ \frac{\underset{(x_0, x_1)\sim\nu}{\mathbb{E}}\left[\exp(f_t) \cdot \partial_t f_t \cdot \gamma\nabla f_t\right]}{\underset{(x_0, x_1)\sim\nu}{\mathbb{E}}\left[\exp(f_t)\right]}$$

$$- \frac{\underset{(x_0, x_1)\sim\nu}{\mathbb{E}}\left[\exp(f_t) \cdot \gamma\nabla f_t\right] \cdot \underset{(x_0, x_1)\sim\nu}{\mathbb{E}}\left[\exp(f_t) \cdot \partial_t f_t\right]}{\left[\underset{(x_0, x_1)\sim\nu}{\mathbb{E}}\left[\exp(f_t)\right]\right]^2}$$

$$= \mathbb{E}[\partial_t(\gamma\nabla f_t)|x_t = x] + \mathrm{Cov}(\gamma\nabla f_t, \partial_t f_t|x_t = x)$$

The above term has exactly the same form as which in the proof of Lemma B.4, so by a similar analysis we can obtain that

$$\mathbb{E}\|\partial_t(\gamma s(t, x_t))\|^2 \leq 2\mathbb{E}\|\partial_t(\gamma\nabla f_t)\|^2 + 2\sqrt{\mathbb{E}|\partial_t f_t|^4} \cdot \sqrt{\mathbb{E}\|\gamma\nabla f_t\|^4}.$$

We have already deduced that
$$\mathbb{E}\|\nabla f_t\|^4 \lesssim \gamma^{-4}d^2,$$

and
$$\mathbb{E}|\partial_t f_t|^4 \lesssim (\gamma^{-1}\dot{\gamma})^4 d^4 + \gamma^{-4}d^2\mathbb{E}\|x_0 - x_1\|^4.$$

Also,
$$\partial_t(\gamma\nabla f_t) = \partial_t\left(-\frac{x - I}{\gamma}\right) = \gamma^{-1}\partial_t I + \gamma^{-2}\dot{\gamma}(x - I) = \gamma^{-1}\partial_t I + \gamma^{-1}\dot{\gamma}z$$

Hence,
$$\mathbb{E}\|\partial_t s(t, x_t)\|^2 \lesssim \gamma^{-2}\dot{\gamma}^2 d^3 + \gamma^{-2}d^2\sqrt{\mathbb{E}\|x_0 - x_1\|^4},$$

which completes the first part. The proof of the second part is exactly the same by replacing $\gamma\nabla f_t$ with $\nabla f_t$. $\qquad\square$

**Lemma B.7.** *For any $p \geq 1$, there exists a constant $C(p) < \infty$ that only depends on $p$ such that*

$$\mathbb{E}\|\nabla s(t, x)\|_F^p \leq C(p)\gamma^{-2p}d^p.$$

*Proof.* With exactly the same ideas of the previous lemmas, we can obtain

$$\nabla s(t, x) = \mathbb{E}[\nabla^2 f_t|x_t = x] + \mathrm{Cov}(\nabla f_t, \nabla f_t|x_t = x)$$
$$= -\gamma^{-2}I + \gamma^{-2}\mathrm{Cov}(z, z|x_t = x)$$

Then, for $p \geq 1$, we have

$$
\begin{aligned}
\mathbb{E}\|\nabla s(t, x_t)\|_F^p &\leq 2^{p-1}\|\gamma^{-2}I\|_F^p + 2^{p-1}\gamma^{-2p}\mathbb{E}\|\mathbb{E}[\|z\|^2 | x_t = x]\|^p \\
&\leq 2^{p-1}\gamma^{-2p}d^{p/2} + 2^{p-1}\gamma^{-2p}\mathbb{E}\|z\|^{2p} \quad\quad \text{(Jensen's inequality)} \\
&\leq C(p)\gamma^{-2p}d^p.
\end{aligned}
$$

Here for the first inequality we have used the fact $(a + b)^p \leq 2^{p-1}a^p + 2^{p-1}b^p$ for $a, b \geq 0$. □

We also need some bounds for $\Delta s$ and $\Delta v$, where $\Delta$ represents the Laplace operator.

**Lemma B.8.**

$$
\mathbb{E}\|\Delta v(t, x_t)\|^2 \lesssim \gamma^{-2}d\mathbb{E}\|x_0 - x_1\|^4 + \gamma^{-4}d^2
$$

for all $t \in (0, 1)$.

*Proof.* We still use the notations in the proof of Lemma B.4. First, in the proof of Lemma B.5, we have already shown that

$$
\begin{aligned}
\partial_{x^i} v(t, x) &= \frac{\underset{(x_0, x_1) \sim \nu}{\mathbb{E}}[\exp(f_t) \cdot (\partial_t I \cdot \partial_{x^i} f_t)]}{\underset{(x_0, x_1) \sim \nu}{\mathbb{E}}[\exp(f_t)]} \\
&\quad - \frac{\underset{(x_0, x_1) \sim \nu}{\mathbb{E}}[\exp(f_t) \cdot \partial_t I] \cdot \underset{(x_0, x_1) \sim \nu}{\mathbb{E}}[\exp(f_t) \cdot \partial_{x^i} f_t]}{\left[\underset{(x_0, x_1) \sim \nu}{\mathbb{E}}[\exp(f_t)]\right]^2} \\
&= \frac{\underset{(x_0, x_1) \sim \nu}{\mathbb{E}}\left[\underset{(\bar{x}_0, \bar{x}_1) \sim \nu}{\mathbb{E}}[\exp(f_t) \exp(\bar{f}_t)(\partial_t I - \partial_t \bar{I}) \cdot (\partial_{x^i} f_t - \partial_{x^i} \bar{f}_t)]\right]}{2\underset{(x_0, x_1) \sim \nu}{\mathbb{E}}\left[\underset{(\bar{x}_0, \bar{x}_1) \sim \nu}{\mathbb{E}}[\exp(f_t) \exp(\bar{f}_t)]\right]}.
\end{aligned}
$$

The last equality is an alternative form of the covariance, and we use notations $\bar{I} = I(t, \bar{x}_0, \bar{x}_1)$ and $\bar{f}_t = f_t(\bar{x}_0, \bar{x}_1)$ for intermediate variables $(\bar{x}_0, \bar{x}_1)$. Hence,

$$
\begin{aligned}
\partial_{x^i}^2 v(t, x) &= \mathrm{Cov}(\partial_t I, \partial_{x^i}^2 f_t | x_t = x) \\
&\quad + \frac{1}{2}\mathrm{Cov}[(\partial_t I - \partial_t \bar{I})(\partial_{x^i} f_t - \partial_{x^i} \bar{f}_t), \partial_{x^i} f_t + \partial_{x^i} \bar{f}_t | x_t = \bar{x}_t = x].
\end{aligned}
$$

For the first term, note that $\partial_{x^i}^2 f_t = -\gamma^{-2}$ is fixed. So,

$$
\Delta v(t, x) = \frac{1}{2}\mathrm{Cov}[(\partial_t I - \partial_t \bar{I})(\nabla f_t - \nabla \bar{f}_t), \nabla f_t + \nabla \bar{f}_t | x_t = \bar{x}_t = x].
$$

Here the covariance refers to the expectation of dot product instead of the expectation of tensor product. Then, use the fact $\mathbb{E}\|X - \mathbb{E}X\|^2 \leq \mathbb{E}\|X\|^2$, we know that

$$
\begin{aligned}
\|\Delta v(t, x)\| &\leq \sqrt{\mathbb{E}[\|(\partial_t I - \partial_t \bar{I})(\nabla f_t - \nabla \bar{f}_t)^T\|^2 | x_t = \bar{x}_t = x]} \\
&\quad \cdot \sqrt{\mathbb{E}[\|\nabla f_t + \nabla \bar{f}_t\|^2 | x_t = \bar{x}_t = x]} \quad\quad \text{(Cauchy-Schwarz inequality)} \\
&\lesssim \left[\mathbb{E}[\|\partial_t I - \partial_t \bar{I}\|^4 | x_t = \bar{x}_t = x]\right]^{1/4} \\
&\quad \cdot \left[\mathbb{E}[\|\nabla f_t - \nabla \bar{f}_t\|^4 | x_t = \bar{x}_t = x]\right]^{1/4} \quad\quad \text{(Cauchy-Schwarz inequality)} \\
&\quad \cdot \sqrt{\mathbb{E}[\|\nabla f_t\|^2 | x_t = x]} \quad\quad\quad\quad\quad\quad\quad\quad \text{(by symmetry)} \\
&\lesssim \left[\mathbb{E}[\|\partial_t I\|^4 | x_t = x]\right]^{1/4} \cdot \sqrt{\mathbb{E}[\|\nabla f_t\|^4 | x_t = x]}. \quad\quad \text{(by symmetry)}
\end{aligned}
$$

Therefore,

$$\mathbb{E}\|\Delta v(t, x_t)\|^2 \lesssim \mathbb{E}\left[\sqrt{\mathbb{E}[\|\partial_t I\|^4 \,|x_t = x]} \cdot \mathbb{E}[\|\nabla f_t\|^4 |x_t = x]\right]$$

$$\lesssim \sqrt{\mathbb{E}\left[\mathbb{E}[\|\partial_t I\|^4 \,|x_t = x]\right]} \cdot \sqrt{\mathbb{E}\left[\mathbb{E}[\|\nabla f_t\|^4 |x_t = x]^2\right]} \quad \text{(Cauchy-Schwarz inequality)}$$

$$\lesssim \gamma^{-4}\sqrt{\mathbb{E}\|\partial_t I\|^4} \cdot \sqrt{\mathbb{E}\|z\|^8} \quad\quad\quad\quad\quad\quad\quad\quad \text{(Jensen's inequality)}$$

$$\lesssim \gamma^{-4}\sqrt{\mathbb{E}\|x_0 - x_1\|^4} \cdot d^2$$

$$\lesssim \gamma^{-2}d\mathbb{E}\|x_0 - x_1\|^4 + \gamma^{-4}d^2.$$

$\square$

**Lemma B.9.**

$$\mathbb{E}\|\Delta s(t, x)\|^2 \lesssim \gamma^{-6}d^3$$

*for $t \in (0, 1)$.*

*Proof.*

$$\nabla s(t, x) = -\gamma^{-2}I + \text{Cov}(\nabla f_t, \nabla f_t | x_t = x).$$

Hence, with similar calculations and notations as in the proof of Lemma B.8, we can deduce that

$$\Delta s(t, x) = 2\text{Cov}(\nabla f_t, \Delta f_t | x_t = x)$$

$$+ \frac{1}{2}\text{Cov}[(\nabla f_t - \nabla \bar{f}_t)(\nabla f_t - \nabla \bar{f}_t)^T, \nabla f_t - \nabla \bar{f}_t | x_t = \bar{x}_t = x].$$

$$= \frac{1}{2}\text{Cov}[(\nabla f_t - \nabla \bar{f}_t)(\nabla f_t - \nabla \bar{f}_t)^T, \nabla f_t - \nabla \bar{f}_t | x_t = \bar{x}_t = x].$$

Then, with Hölder's inequality, we have

$$\|\Delta s(t, x)\| \lesssim \left[\mathbb{E}[\|\nabla f_t - \nabla \bar{f}_t\|^3 | x_t = \bar{x}_t = x]\right]^{1/3}$$

$$\cdot \left[\mathbb{E}[\|(\nabla f_t - \nabla \bar{f}_t)(\nabla f_t - \nabla \bar{f}_t)^T\|^{3/2}]\right]^{2/3}$$

$$\lesssim \left[\mathbb{E}[\|\nabla f_t\|^3 | x_t = x]\right]^{1/3} \quad\quad\quad\quad\quad \text{(by symmetry)}$$

$$\cdot \left[\mathbb{E}[\|\nabla f_t - \nabla \bar{f}_t\|^3 | x_t = \bar{x}_t = x]\right]^{2/3}$$

$$\lesssim \mathbb{E}[\|\nabla f_t\|^3 | x_t = x]. \quad\quad\quad\quad\quad\quad \text{(by symmetry)}$$

Hence, by Jensen's inequality,

$$\mathbb{E}\|\Delta s(t, x)\|^2 \lesssim \mathbb{E}[\mathbb{E}[\|\nabla f_t\|^3 | x_t = x]^2] \lesssim \gamma^{-6}\mathbb{E}\|z\|^6 \lesssim \gamma^{-6}d^3.$$

$\square$

## C. Omitted Proofs in Sections 4 and 5

### C.1. Bounds along the forward Path

Recall the forward ODE

$$\mathrm{d}X_t = b(t, X_t)\mathrm{d}t$$

and the forward SDE

$$\mathrm{d}X_t^F = b_F(t, X_t^F)\mathrm{d}t + \sqrt{2\epsilon}dW_t.$$

Their solutions are denoted by $X_t$ and $X_t^F$, respectively. Using the chain rule or Itô's formula, for a function $f(t, x)$ that is twice continuously differentiable, we have

$$\mathrm{d}f(t, X_t) = [\partial_t f(t, X_t) + \nabla f(t, X_t) \cdot b(t, X_t)]\mathrm{d}t,$$

and

$$df(t, X_t^F) = [\partial_t f(t, X_t^F) + \nabla f(t, X_t^F) \cdot b_F(t, X_t^F) + \epsilon \Delta f]dt + \sqrt{2\epsilon}\nabla f(t, X_t^F) \cdot dW_t.$$

With the above formula, we can now provide the following bound on the discretization error.

**Lemma C.1.** *For $0 < t_0 \leq t_1 < 1$, suppose $\epsilon = O(1)$, then,*

$$\mathbb{E}\|v(t_0, X_{t_0}^F) - v(t_1, X_{t_1}^F)\|^2 \lesssim (t_1 - t_0)^2 \left[ M_2 + \gamma_{\min}^{-6}d^3 + \gamma_{\min}^{-2}d\sqrt{\mathbb{E}\|x_0 - x_1\|^8} \right]$$
$$+ \epsilon(t_1 - t_0)\gamma_{\min}^{-2}d\sqrt{\mathbb{E}\|x_0 - x_1\|^4},$$

*and*

$$\mathbb{E}\|s(t_0, X_{t_0}^F) - s(t_1, X_{t_1}^F)\|^2 \lesssim (t_1 - t_0)^2 \left[ \gamma_{\min}^{-4}d^2\sqrt{\mathbb{E}\|x_0 - x_1\|^4} + \gamma_{\min}^{-6}d^3 \right] + \epsilon(t_1 - t_0)\gamma_{\min}^{-4}d^2.$$

*Here we denote $\gamma_{\min} = \min_{u \in [t_0, t_1]} \gamma$.*

*Proof.* According to the formula

$$dv(t, X_t^F) = [\partial_t v(t, X_t^F) + \nabla v(t, X_t^F) \cdot b_F(t, X_t^F) + \epsilon \Delta v(t, X_t^F)]dt + \sqrt{2\epsilon}\nabla v(t, X_t^F) \cdot dW_t,$$

we know that

$$\|v(t_1, X_{t_1}^F) - v(t_0, X_{t_0}^F)\|^2 \leq 4\left\| \int_{t_0}^{t_1} \partial_u v(u, X_u^F)du \right\|^2$$
$$+ 4\left\| \int_{t_0}^{t_1} \nabla v(u, X_u^F) \cdot b(u, X_u^F)du \right\|^2$$
$$+ 4\left\| \int_{t_0}^{t_1} \epsilon \Delta v(u, X_u^F)du \right\|^2$$
$$+ 4\left\| \int_{t_0}^{t_1} \sqrt{2\epsilon}\nabla v(u, X_u^F) \cdot dW_u \right\|^2.$$

For the first three terms, by Jensen's inequality we know that for any function $Y$, we have

$$\left\| \int_{t_0}^{t_1} Y(u)du \right\|^2 \leq (t_1 - t_0) \int_{t_0}^{t_1} \|Y(u)\|^2 du.$$

For the last term, use Itô's isometry (Le Gall 2016, Equation 5.8), we can get

$$\mathbb{E}\left[ \left\| \int_{t_0}^{t_1} \sqrt{2\epsilon}\nabla v(u, X_u^F) \cdot dW_u \right\|^2 \right] = \int_{t_0}^{t_1} \mathbb{E}\|\sqrt{2\epsilon}\nabla v(u, X_u^F)\|_F^2 du.$$

Therefore, we can use Fubini's theorem to change the order of expectation and integral, and combine the results of Lemma B.4, B.5, B.8 and B.3 and Assumption 4.1 to get

$$\|v(t_1, X_{t_1}^F) - v(t_0, X_{t_0}^F)\|^2 \lesssim (t_1 - t_0) \int_{t_0}^{t_1} \mathbb{E}\|\partial_u v(u, X_u)\|^2 du$$
$$+ (t_1 - t_0) \int_{t_0}^{t_1} \left[ \gamma_{\min}^2 d^{-1}\mathbb{E}\|\nabla v(u, X_u)\|^4 + \gamma_{\min}^{-2}d\mathbb{E}\|b_F(u, X_u)\|^4 \right] du$$
$$+ \epsilon^2(t_1 - t_0) \int_{t_0}^{t_1} \mathbb{E}\|\Delta v(u, X_u)\|^2 du$$
$$+ 2\epsilon \int_{t_0}^{t_1} \mathbb{E}\|\nabla v(u, X_u^F)\|_F^2 du$$
$$\lesssim (t_1 - t_0)^2 \left[ M_2 + \gamma_{\min}^{-6}d^3 + \gamma_{\min}^{-2}d\sqrt{\mathbb{E}\|x_0 - x_1\|^8} \right]$$
$$+ \epsilon(t_1 - t_0)\gamma^{-2}d^1\sqrt{\mathbb{E}\|x_0 - x_1\|^4}$$

Note that we have already used the condition $\gamma^2 \in C^2[0,1]$ and $\gamma\dot{\gamma} = O(1)$.

Similarly, we can use Lemma B.6, B.7, B.9 and B.3, and the formula

$$\mathrm{d}s(t, X_t^F) = [\partial_t s(t, X_t^F) + \nabla s(t, X_t^F) \cdot b_F(t, X_t^F) + \epsilon \Delta s(t, X_t^F)]\mathrm{d}t + \sqrt{2\epsilon}\nabla s(t, X_t^F) \cdot \mathrm{d}W_t$$

to bound

$$\mathbb{E}\|s(t_1, X_{t_1}^F) - s(t_0, X_{t_0}^F)\|^2 \lesssim (t_1 - t_0)^2 \left[\gamma_{\min}^{-4} d^2 \sqrt{\mathbb{E}\|x_0 - x_1\|^4} + \gamma_{\min}^{-6} d^3\right] + \epsilon(t_1 - t_0)\gamma_{\min}^{-4} d^2,$$

where $\gamma_{\min} = \min_{u \in [s,t]} \gamma$. $\qquad\qquad\square$

## C.2. Proof of Theorem 4.3

We first give the following proposition, which is a result from (Chen et al., 2023d).

**Proposition C.2.** *(Section 5.2 of (Chen et al., 2023d)) Let $P, Q$ be the path measures of solutions of SDE (3) and (6), where they both start from the same distribution $\rho(t_0)$ at time $t = t_0$ and end at time $t = t_N$. Then, if*

$$\mathbb{E}[\|b_F(t, X_t^F) - \hat{b}_F(t_k, X_{t_k}^F)\|^2] \leq C$$

*for any $t \in [t_0, t_N]$ and some constant $C$, we have*

$$KL(P\|Q) = \frac{1}{4\epsilon}\sum_{k=0}^{N-1}\int_{t_k}^{t_{k+1}} \mathbb{E}[\|b_F(t, X_t^F) - \hat{b}_F(t_k, X_{t_k}^F)\|^2]\mathrm{d}t.$$

*Here the expectations are taken over the ground-truth forward process $(X_t^F)_{t\in[t_0,t_N]} \sim P$.*

Now, using the above proposition, we are ready to prove Theorem 4.3.

*Proof.* Let $P, Q$ be the path measures of the solutions to the SDE (3) and (6), where the solutions start from the same distribution $\rho(t_0)$ at time $t = t_0$, as in Proposition C.2. We first want to check the condition of Proposition C.2. Note that

$$
\begin{aligned}
\mathbb{E}\|\hat{b}_F(t_k, X_{t_k}^F) - b_F(t, X_t^F)\|^2 \overset{(a)}{\leq}\ & 2\mathbb{E}\|\hat{b}_F(t_k, X_{t_k}^F) - b_F(t_k, X_{t_k}^F)\|^2 + 2\mathbb{E}\|b_F(t_k, X_{t_k}^F) - b_F(t, X_t^F)\|^2 \\
\overset{(b)}{\leq}\ & 2\mathbb{E}\|\hat{b}_F(t_k, X_{t_k}^F) - b_F(t_k, X_{t_k}^F)\|^2 + 4\mathbb{E}\|v(t_k, X_{t_k}^F) - v(t, X_t^F)\|^2 \\
& + 4\mathbb{E}\|(-\gamma(t_k)\dot{\gamma}(t_k) + \epsilon)s(t_k, X_{t_k}^F) - (-\gamma(t_k)\dot{\gamma}(t_k) + \epsilon)s(t, X_t^F)\|^2 \\
\overset{(c)}{\leq}\ & 2\mathbb{E}\|\hat{b}_F(t_k, X_{t_k}^F) - b_F(t_k, X_{t_k}^F)\|^2 + 4\mathbb{E}\|v(t_k, X_{t_k}^F) - v(t, X_t^F)\|^2 \\
& + 8(-\gamma(t_k)\dot{\gamma}(t_k) + \epsilon)^2\mathbb{E}\|s(t_k, X_{t_k}^F) - s(t, X_t^F)\|^2 \\
& + 8(\gamma(t)\dot{\gamma}(t) - \gamma(t_k)\dot{\gamma}(t_k))^2\mathbb{E}\|s(t, X_t^F)\|^2.
\end{aligned}
$$

Here (a), (b) and (c) use the triangle inequality and the fact $(a + b)^2 \leq 2a^2 + 2b^2$. By Lemmas B.3 and C.1, this term is uniformly bounded in the closed interval $[t_0, t_N]$. In fact, we can apply these lemmas to obtain that

$$
\begin{aligned}
\mathbb{E}\|\hat{b}_F(t_k, X_{t_k}^F) - b_F(t, X_t^F)\|^2 \overset{(a)}{\lesssim}\ & \mathbb{E}\|\hat{b}_F(t_k, X_{t_k}^F) - b_F(t_k, X_{t_k}^F)\|^2 \\
& + (t - t_k)^2 \left[M_2 + \bar{\gamma}_k^{-6} d^3 + \bar{\gamma}_k^{-2} d\sqrt{\mathbb{E}\|x_0 - x_1\|^8}\right] \\
& + \epsilon(t - t_k)\bar{\gamma}_k^{-2} d\sqrt{\mathbb{E}\|x_0 - x_1\|^4} \\
& + (t - t_k)^2 \left[\bar{\gamma}_k^{-4} d^2\sqrt{\mathbb{E}\|x_0 - x_1\|^4} + \bar{\gamma}_k^{-6} d^3\right] + \epsilon(t - t_k)\bar{\gamma}_k^{-4} d^2 \\
& + (t - t_k)^2\bar{\gamma}_k^{-2} d \\
\overset{(b)}{\lesssim}\ & \mathbb{E}\|\hat{b}_F(t_k, X_{t_k}^F) - b_F(t_k, X_{t_k}^F)\|^2 \\
& + (t - t_k)^2 \left[M_2 + \bar{\gamma}_k^{-6} d^3 + \bar{\gamma}_k^{-2} d\sqrt{\mathbb{E}\|x_0 - x_1\|^8}\right] \\
& + \epsilon(t - t_k)\bar{\gamma}_k^{-2} d\left[\sqrt{\mathbb{E}\|x_0 - x_1\|^4} + \bar{\gamma}_k^{-2} d\right].
\end{aligned}
$$

Here step (a) directly expands the discretization error using Lemmas B.3 and C.1; step (b) simplifies the terms by applying Young's inequality and that $1 + \epsilon^2 = O(1)$. Then, by Proposition C.2,

$$\mathrm{KL}(P\|Q) = \frac{1}{4\epsilon} \sum_{k=0}^{N-1} \int_{t_k}^{t_{k+1}} \mathbb{E}[\|b_F(t, X_t^F) - \hat{b}_F(t_k, X_{t_k}^F)\|^2]\mathrm{d}t$$

$$\overset{(a)}{\lesssim} \epsilon^{-1}\varepsilon_{b_F}^2 + \epsilon^{-1} \sum_{k=0}^{N-1} (t_{k+1} - t_k)^3 \left[ M_2 + \bar{\gamma}_k^{-6}d^3 + \bar{\gamma}_k^{-2}d\sqrt{\mathbb{E}\|x_0 - x_1\|^8} \right]$$

$$+ \sum_{k=0}^{N-1} (t_{k+1} - t_k)^2\bar{\gamma}_k^{-2}d \left[ \sqrt{\mathbb{E}\|x_0 - x_1\|^4} + \bar{\gamma}_k^{-2}d \right].$$

Here step (a) just integrates over the upper bound of the disretization error. Now, consider $\mathrm{KL}(\rho(t_N)\|\hat{\rho}(t_N))$. Let $\hat{Q}$ be the path measure of solutions of (6) starting from $\hat{\rho}(t_0)$ instead of $\rho(t_0)$. Then,

$$\mathrm{KL}(\rho(t_N)\|\hat{\rho}(t_N)) \leq \mathrm{KL}(P\|\hat{Q}) = \mathbb{E}_P\left[\log\frac{\mathrm{d}P}{\mathrm{d}\hat{Q}}(X)\right]$$

$$= \mathbb{E}_P\left[\log\left(\frac{\mathrm{d}P}{\mathrm{d}Q}(X) \cdot \frac{\mathrm{d}Q}{\mathrm{d}\hat{Q}}(X)\right)\right]$$

$$= \mathbb{E}_P\left[\log\frac{\mathrm{d}P}{\mathrm{d}Q}(X)\right] + \mathbb{E}_P\left[\log\frac{\mathrm{d}\rho(t_0)}{\mathrm{d}\hat{\rho}(t_0)}(X_{t_0})\right]$$

$$= \mathrm{KL}(P\|Q) + \mathrm{KL}(\rho(t_0)\|\hat{\rho}(t_0)).$$

The proof is then completed. $\qquad\square$

### C.3. Proof of Proposition 5.1

*Proof.* Using the results of Theorem 4.3,

$$\mathrm{KL}(\rho(t_N)\|\hat{\rho}(t_N)) \overset{(a)}{\lesssim} \varepsilon_{b_F}^2 + \mathrm{KL}(\rho(t_0)\|\hat{\rho}(t_0)) + \epsilon^{-1} \sum_{k=0}^{N-1} (t_{k+1} - t_k)^3 \left[ M_2 + \bar{\gamma}_k^{-6}d^3 + \bar{\gamma}_k^{-2}d\sqrt{\mathbb{E}\|x_0 - x_1\|^8} \right]$$

$$+ \sum_{k=0}^{N-1} (t_{k+1} - t_k)^2\bar{\gamma}_k^{-2}d \left[ \sqrt{\mathbb{E}\|x_0 - x_1\|^4} + \bar{\gamma}_k^{-2}d \right]$$

$$\overset{(b)}{\lesssim} \varepsilon_{b_F}^2 + \mathrm{KL}(\rho(t_0)\|\hat{\rho}(t_0)) + \epsilon^{-1} \sum_{k=0}^{N-1} \left[ M_2 h_k^3 + h^3 d^3 + h h_k^2 d\sqrt{\mathbb{E}\|x_0 - x_1\|^8} \right]$$

$$+ \sum_{k=0}^{N-1} \left[ h h_k d\sqrt{\mathbb{E}\|x_0 - x_1\|^4} + h^2 d^2 \right]$$

$$\overset{(c)}{\lesssim} \varepsilon_{b_F}^2 + \mathrm{KL}(\rho(t_0)\|\hat{\rho}(t_0)) + \epsilon^{-1}h^2 \left( M_2 + d\sqrt{\mathbb{E}\|x_0 - x_1\|^8} \right) + \epsilon^{-1}Nh^3 d^3$$

$$+ hd\sqrt{\mathbb{E}\|x_0 - x_1\|^4} + Nh^2 d^2$$

$$\overset{(d)}{\lesssim} \varepsilon_{b_F}^2 + \mathrm{KL}(\rho(t_0)\|\hat{\rho}(t_0)) + hd\sqrt{\mathbb{E}\|x_0 - x_1\|^4} + Nh^2 d^2.$$

Here step (a) is the result of Theorem 4.3; step (b) uses the fact $h_k = t_{k+1} - t_k = O(h\bar{\gamma}_k^2)$; step (c) uses the fact $\sum_{k=0}^{N-1} h_k = t_N - t_0 \leq 1$; step (d) omits the higher-order terms. $\qquad\square$

### C.4. Proof of Corollary 5.2

*Proof.* When the number of steps is $N$, we have

$$h = \Theta\left( N^{-1} \log\left(\frac{1}{t_0(1 - t_N)}\right) \right).$$

Then, by Corollary 5.1 and the assumptions,

$$\mathrm{KL}(\rho(t_N)\|\hat{\rho}(t_N)) \lesssim \varepsilon^2 + N^{-1} \left[ d\sqrt{\mathbb{E}\|x_0 - x_1\|^4} \log\left(\frac{1}{t_0(1 - t_N)}\right) + d^2 \log^2\left(\frac{1}{t_0(1 - t_N)}\right) \right].$$

This gives the complexity to to make $\mathrm{KL}(\rho(t_N)\|\hat{\rho}(t_N)) \lesssim \varepsilon^2$. $\qquad\square$

### C.5. Reducing to Diffusion Models

By modifying the definition of stochastic interpolant to

$$x_t = I(t, x_1) + \gamma(t)z$$

and change the condition on $I$ to $\|\partial_t I(t, x_1)\| \leq C\|x_1\|$, we can repeat the previous analysis while replacing $\sqrt{\mathbb{E}\|x_0 - x_1\|^p}$ by $\sqrt{\mathbb{E}\|x_1\|^p}$. For the case of diffusion models, we can choose $I(t, x_1) = tx_1$ and $\gamma(t) = \sqrt{1 - t^2}$ to obtain a process with the same marginal distributions. Moreover, under this definition of interpolants, we can choose $t_0 = 0$ and $h_k = t_{k+1} - t_k \propto (1 - t_k)$ as the time schedule to recover the sample complexity of diffusion models.

### C.6. Omitted Proofs for $\gamma^2(t) = (1 - t)^2 t$

In this section, we will design a schedule for $\gamma^2(t) = (1 - t)^2 t$, and provide the corresponding complexity deduced using Theorem 4.3. Moreover, we also derived the complexity of using a uniform schedule for comparison.

**Corollary C.3.** *For $\gamma^2(t) = (1 - t)^2 t$, there exists a schedule so that under the same assumptions as Corollary 5.2, the complexity is given by*

$$N = O\left(\frac{1}{\varepsilon^2} \left[ \sqrt{\mathbb{E}\|x_0 - x_1\|^4} d\left(\frac{1}{\sqrt{1 - t_N}} + \log\left(\frac{1}{t_0}\right)\right) + d^2\left(\frac{1}{(1 - t_N)^2} + \log^2\left(\frac{1}{t_0}\right)\right) \right]\right).$$

*In addition, the complexity for using a uniform schedule is*

$$N = O\left(\frac{1}{\varepsilon^2} \left[ \sqrt{\mathbb{E}\|x_0 - x_1\|^4} d\left(\frac{1}{1 - t_N} + \log\left(\frac{1}{t_0}\right)\right) + d^2\left(\frac{1}{(1 - t_N)^3} + \frac{1}{t_0}\right) \right]\right).$$

*Proof.* Here we also take $h_M = 0.5$ for some $M > 0$. Then, we define

$$h_k = \begin{cases} h_A \cdot t_{k+1}, & k < M \\ h_B \cdot (1 - t_k)^{1.5}, & k \geq M. \end{cases}$$

for some $h_A \in [0, 0.5), h_B \in [0, 1)$. For the part $k < M$ and $t_k \in [0, 0.5)$, $\gamma^2(t_k) = \Theta(t_k)$, so it is the same as what we have discussed for the case, and we need

$$M = N_1 = O\left(\frac{1}{\varepsilon^2} \left[ \sqrt{\mathbb{E}\|x_0 - x_1\|^4} d \log\left(\frac{1}{t_0}\right) + d^2 \log^2\left(\frac{1}{t_0}\right) \right]\right)$$

steps to make the discretization error

$$\varepsilon_{b_F}^2 + \sum_{k=0}^{M-1} (t_{k+1} - t_k)^3 \left[ M_2 + \bar{\gamma}_k^{-6} d^3 + \bar{\gamma}_k^{-2} d\sqrt{\mathbb{E}\|x_0 - x_1\|^8} \right]$$

$$+ \sum_{k=0}^{M-1} (t_{k+1} - t_k)^2 \bar{\gamma}_k^{-2} d \left[ \sqrt{\mathbb{E}\|x_0 - x_1\|^4} + \bar{\gamma}_k^{-2} d \right] \lesssim \varepsilon^2.$$

For the part $k \geq M$,

$$\sum_{k=M}^{N-1} (t_{k+1} - t_k)^3 \left[ M_2 + \bar{\gamma}_k^{-6} d^3 + \bar{\gamma}_k^{-2} d\sqrt{\mathbb{E}\|x_0 - x_1\|^8} \right] = O(h_B^2),$$

and by that $h_k = \Theta(h_B \bar{\gamma}_k^{1.5}) = \Theta(h_B(1 - t_k)^{1.5})$ (use in step (a) below),

$$\sum_{k=M}^{N-1} h_k^2 \bar{\gamma}_k^{-2} d \left[ \sqrt{\mathbb{E}\|x_0 - x_1\|^4} + \bar{\gamma}_k^{-2} d \right]$$

$$\overset{(a)}{\lesssim} h_B \sum_{k=M}^{N-1} h_k \left[ d\sqrt{\mathbb{E}\|x_0 - x_1\|^4} \bar{\gamma}_k^{-0.5} + \bar{\gamma}_k^{-2.5} d^2 \right]$$

$$\overset{(b)}{\lesssim} h_B \left[ d\sqrt{\mathbb{E}\|x_0 - x_1\|^4} \int_{0.5}^{t_N} (1 - s)^{-0.5} \mathrm{d}s + d^2 \int_{0.5}^{t_N} (1 - s)^{-2.5} \mathrm{d}s \right]$$

$$\lesssim h_B \left[ d\sqrt{\mathbb{E}\|x_0 - x_1\|^4} + d^2 \frac{1}{(1 - t_N)^{1.5}} \right].$$

Here the inequality (b) is by that $\bar{\gamma}_k = \Theta(1 - t)$ for $t \in [t_k, t_{k+1}]$. Now, we want to compute the number of steps $N_2 = N - M$ for the part $k \geq M$. Note that if $t_k = 1 - 2^{-p}$, it takes $O(2^{p/2} h_B^{-1})$ more steps to reach $1 - 2^{-p-1}$. Hence $N_2 = O\left(h_B^{-1}(1 - t_N)^{-0.5}\right)$, so we need to take $h_B = \Theta\left(N^{-1}(1 - t_N)^{-0.5}\right)$. Therefore,

$$\sum_{k=M}^{N-1} h_k^2 \bar{\gamma}_k^{-2} d \left[ \sqrt{\mathbb{E}\|x_0 - x_1\|^4} + \bar{\gamma}_k^{-2} d \right]$$

$$\lesssim N^{-1} \left[ \frac{d\sqrt{\mathbb{E}\|x_0 - x_1\|^4}}{\sqrt{1 - t_N}} + \frac{d^2}{(1 - t_N)^2} \right].$$

Thus, for the part $k > M$, we need

$$N - M = N_2 = O\left( \frac{1}{\varepsilon^2} \left( \frac{d\sqrt{\mathbb{E}\|x_0 - x_1\|^4}}{\sqrt{1 - t_N}} + \frac{d^2}{(1 - t_N)^2} \right) \right)$$

steps to make the discretization error bounded by $O(\varepsilon^2)$. Hence, the overall complexity is given by $N = N_1 + N_2$, which is our result.

If we use a uniform schedule, by Theorem 4.3 and that $\gamma^2(t) = \Theta(\min\{t, (1 - t)^2\})$, we can bound

$$\mathrm{KL}(\rho(t_N)\|\hat{\rho}(t_N)) \overset{(a)}{\lesssim} \varepsilon_{b_F}^2 + \mathrm{KL}(\rho(t_0)\|\hat{\rho}(t_0))$$

$$+ \frac{1}{N}\sqrt{\mathbb{E}\|x_0 - x_1\|^4} d \left( \int_{t_0}^{0.5} s^{-1} \mathrm{d}s + \int_{0.5}^{t_N} (1 - s)^{-2} \mathrm{d}s \right)$$

$$+ \frac{1}{N} d^2 \left( \int_{t_0}^{0.5} s^{-2} \mathrm{d}s + \int_{0.5}^{t_N} (1 - s)^{-4} \mathrm{d}s \right)$$

$$\lesssim \varepsilon_{b_F}^2 + \mathrm{KL}(\rho(t_0)\|\hat{\rho}(t_0))$$

$$+ \frac{1}{N}\sqrt{\mathbb{E}\|x_0 - x_1\|^4} d \left( \frac{1}{1 - t_N} + \log\left(\frac{1}{t_0}\right) \right)$$

$$+ \frac{1}{N} d^2 \left( \frac{1}{(1 - t_N)^3} + \frac{1}{t_0} \right),$$

which further gives the complexity bound for uniform schedule. Here the inequality (a) is by applying Theorem 4.3 and replacing $\bar{\gamma}_k$ with the term of the same order. This bound is worse than using the schedule satisfying that $h_k \lesssim h\bar{\gamma}_k$. $\qquad\square$

## D. More Details of Numerical Experiments

To parameterize the estimator $\hat{b}_F(t, x)$ for two-dimensional data, we utilize a simple multilayer perceptron (MLP) network. The input of the network comprises a three-dimensional vector $(x, t)$, and its output is a two-dimensional vector $\hat{b}_F(t, x)$. The MLP architecture consists of three hidden layers, each with 256 neurons, followed by ReLU activation functions (Nair & Hinton, 2010).

To train the estimator $\hat{b}_F(t, x)$, we leverage a simple quadratic objective (see Appendix A for details) whose optimizer is the real drift $b_F(t, x)$. Given the estimator, data batches, and sampled time points, we are ready to compute an empirical loss. We employ the Adam optimizer (Kingma & Ba, 2015) to train the network using the gradient computed on the empirical loss.

We set $t_0 = 0.001$ and $t_N = 0.999$ to ensure that the initial density $\rho(t_0)$ is close to $\rho_0$ and the estimated density $\rho(t_N)$ closely approximates $\rho_1$. During the training process of the estimator $\hat{b}_F(t, x)$, we employ an importance sampling technique so that $t \in [t_0, t_N]$ is sampled with probability proportional to $\gamma^{-2}(t)$. We implement the discretized sampler as defined in Equation (7). We use more than $10,000$ data samples to empirically visualize the densities in Figures 1, 2a and 2b.

In addition, for KL divergence estimation, we estimate the density ratio by comparing the distance to the $k$-th nearest neighborhood. Specifically, for some $x \in \mathbb{R}^d$, let $\{x_0^i\}_{i=1}^n$ and $\{x_1^j\}_{j=0}^m$ be i.i.d. samples from $\rho_0$ and $\rho_1$, and $d_0, d_1$ be the corresponding distance from $x$ to the $k$-th nearest neighborhood, then we estimate

$$\frac{\rho_0(x)}{\rho_1(x)} \approx \frac{k/(n d_0^d)}{k/(m d_1^d)} = \frac{m d_1^d}{n d_0^d}.$$

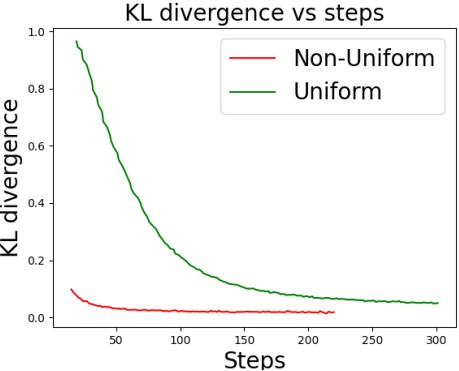

*Figure 5.* Estimated KL divergence for different step size schedules, where we use $\gamma^2(t) = (1-t)^2 t$. The red curve denotes the distance when we use the schedule designed in Appendix C.6, while the green curve denotes the distance when we use the uniform schedule.

## D.1. Additional Experiments for $\gamma(t) = \sqrt{(1-t)^2 t}$

We implement the schedule discussed in Appendix C.6 and compare it to the uniform schedule. We choose $(t_0, t_N) = (0.001, 0.97)$ since $\gamma^2(t)$ is $\Theta((1-t)^2)$ near $t = 1$. We choose $\rho_0$ as the "checkerboard" density and $\rho_1$ as the "spiral" density. We estimate the KL divergence $\text{KL}(\rho(t_N) \| \hat{\rho}(t_N))$ to indicate how close the estimated distribution is to the target distribution. The comparison is shown in Figure 5.

