# OpenReview forum: "Finite-Time Analysis of Discrete-Time Stochastic Interpolants"
_ICML.cc/2025/Conference — ICML 2025 poster_

### Official Review · Reviewer_NVAq · 2025-02-14

**Overall Recommendation:** 4

**Summary:**

This work provides a theoretical analysis of time-discretized stochastic interpolant models. Specifically, they address the problem of convergence with respect to the number of steps in the time discretization, characterizing the error in the modeled distribution with respect to the discretization scheme, the model approximation error, and the prior approximation error. This extends prior analysis focusing solely on diffusion models while remaining consistent, and addresses the effect of time discretization in particular. Motivated by their theoretical results, the authors outline a new time discretization strategy which can theoretically improve convergence of discrete-time solvers using stochastic interpolants, which they demonstrate empirically.

## update after rebuttal: I believe my original score holds, as this remains a strong paper and contribution.

**Claims And Evidence:**

The primary claim made by the authors concerns the convergence of discrete-time solvers (namely, Euler-Maruyama) for stochastic interpolant models. They provide a bound on the KL divergence between the true and model target distributions, which is a function of the model approximation error, time discretization error, and prior approximation “distance”. The proof of this method is explained intuitively, with a detailed proof provided in the Appendix. This bound is then used to design an efficient time discretization schedule to ensure more rapid convergence with respect to the number of time steps. They show theoretically and empirically that their new strategy yields more efficient sampling.

**Essential References Not Discussed:**

I do not believe there is any missing relevant literature.

**Experimental Designs Or Analyses:**

As mentioned above, I believe their limited experiments provide sufficient empirical evidence that their theory is sound. While the level of experimental evaluation is not the same as a typical practical generative modeling paper, I believe their experiments do well to demonstrate the validity of their theoretical results.

**Methods And Evaluation Criteria:**

While the primary focus of this work is a theoretical characterization of practical methods for computing stochastic interpolants, the authors also provide limited experimental validation of their claims. I find this evaluation very convincing, as they demonstrate the enhanced performance of their scheduling scheme as predicted by their theory. While this is demonstrated on toy datasets, the point is made very clearly.

**Other Comments Or Suggestions:**

Right Column Line 170: “initial distribution mismatch  (i.e., KL( ρ(t0)||ρ^(tN) ))” it should be KL( ρ(t0)||ρ^(t0) )?

Right Column Lines 361-365: There seems to be a grammatical error here, part of the sentence is repeated.

Right Column Line 373: aspeccts - > aspects

**Other Strengths And Weaknesses:**

Overall, I believe this would be a very valuable contribution to the community. The authors do well to highlight the importance of considering time discretization in addition to model approximation error in modern generative modeling convergence/error analysis. Moreover, their theoretical insights provide immediate practical considerations that can be employed in stochastic interpolant samplers, as they demonstrate both theoretically and empirically that their proposed sampling scheme can yield better convergence than a simple scheme used in practice, such as uniform discretization. They provide a theoretical basis for designing SDE-based sampling schemes targeting specific error bounds.

**Questions For Authors:**

Is there some way to selected or design \gamma to yield efficient sampling?

**Relation To Broader Scientific Literature:**

The authors position their work within two main areas in the literature. First, their theoretical analysis focuses on stochastic interpolants related to continuous-time normalizing flows and diffusion. Such approaches have seen widespread application in high-dimensional, structured generative modeling settings (e.g., image generation). Second, they provide an analogous study of stochastic interpolants to those conducted for the convergence and error analysis of diffusion.

**Theoretical Claims:**

The authors provide intuitive interpretations and walkthroughs of their theorems and proofs. I found these descriptions intuitive and easy to follow, and they are convincingly backed by rigorous proof.

---

> ### Author Rebuttal · Authors · 2025-03-31
>
> Thank you for your time and effort in reviewing our paper! We are grateful for your constructive suggestions, which have significantly guided our improvements. Please find our responses to your comments below.
>
> ### Other Comments Or Suggestions:
>
> > Right Column Line 170: “initial distribution mismatch (i.e., $\text{KL}( \rho(t_0)\Vert\hat{\rho}(t_N))$)” it should be $\text{KL}( \rho(t_0)\Vert\hat{\rho}(t_0))$?\
> > Right Column Lines 361-365: There seems to be a grammatical error here, part of the sentence is repeated.\
> > Right Column Line 373: aspeccts - > aspects
>
> A: Thanks for your suggestion! We will update them in the final version.
>
> ### Questions For Authors:
>
> > Is there some way to selected or design $\gamma$ to yield efficient sampling?
>
> A: Thanks for your insightful question! Our current analysis regarding schedule design and its associated sample complexity is predicated on a fixed latent scale, $\gamma$. In Section 5, we adopt $\gamma(t)=\sqrt{at(1-t)}$ due to its prevalence and natural appeal, stemming from its connection to the Brownian bridge process. However, the systematic design of $\gamma$ to optimize sampling efficiency represents a compelling avenue for future research, potentially necessitating a synergistic approach encompassing both theoretical analysis and practical experimentation.
>
> ---
> We hope our response addresses your concerns. If so, we wonder if you could kindly consider raising your score? We will also be happy to answer any further questions you may have. Thank you very much!

---

> > ### Comment · Reviewer_NVAq · 2025-04-05
> >
> > Thank you for addressing my limited concerns/questions. I believe my original score holds, and that this work represents a useful contribution.

---

### Official Review · Reviewer_s1b3 · 2025-03-13

**Overall Recommendation:** 3

**Summary:**

This paper present discrete-time analysis of the stochastic interpolant framework or as known as flow models, by theorical results, this paper design a schedules for convergence acceleration.

## update after rebuttal

My view has not changed, so I maintain my original score.

**Claims And Evidence:**

I think the discrete-time analysis is reasonable.

**Essential References Not Discussed:**

Flow straight and fast: Learning to generate and transfer data with rectified flow ICLR 2023
Flow matching for generative modeling ICLR 2023

**Experimental Designs Or Analyses:**

1.	Although the authors provide theoretical justification for their method’s superiority, the experiments themselves appear overly simple. I recommend conducting toy experiments on low-resolution datasets such as CIFAR-10 or ImageNet32 since training flow models on these datasets are relatively fast and computationally manageable.
2.	Experiments conducted solely on Gaussian data only confirm the theoretical foundations. Given that the authors’ theory largely transfers discrete-time analysis to flow models, it would be more convincing to present additional experiments that demonstrate the claimed ability to “design efficient schedules for convergence acceleration.”

**Methods And Evaluation Criteria:**

1.	I believe the authors should explain the novelty of the Exponentially Decaying Time Schedule. Given that flow models and diffusion models are not substantially different, it appears that the authors may have merely transferred a time schedule from diffusion to flow models.
2.	If that is the case, the main contribution seems to be theoretically verifying that the Exponentially Decaying Time Schedule offers certain advantages over a Uniform Schedule in flow models.

**Other Comments Or Suggestions:**

I believe the authors’ statement that the paper focuses on theory is valid. However, if they aim to provide an improved method, they should at least include standard experiments on datasets like CIFAR-10 to support their claims. Relying solely on simple two-dimensional datasets is not sufficient.

**Other Strengths And Weaknesses:**

No

**Questions For Authors:**

See above.

**Relation To Broader Scientific Literature:**

This paper relates to many applications of diffusion models.

**Theoretical Claims:**

Theoretical claim is valid.

---

> ### Author Rebuttal · Authors · 2025-03-31
>
> Thank you for your time and effort in reviewing our paper! We are grateful for your constructive suggestions, which have significantly guided our improvements. Please find our responses to your comments below.
> ### Methods And Evaluation Criteria
>
> > I believe the authors should explain the novelty of the Exponentially Decaying Time Schedule. Given that flow models and diffusion models are not substantially different, it appears that the authors may have merely transferred a time schedule from diffusion to flow models.
>
> A: We first would like to emphasize that our work introduces the first discrete-time convergence bound for SDE-based generative models within the stochastic interpolants framework. Theorem 4.3 provides an explicit upper bound on the estimation error, expressed in terms of step sizes $h_k$, dimension $d$, latent scale $\gamma$, and the distance between two distributions measured by $\mathbb{E}\Vert x_0-x_1\Vert^p$. This theorem establishes an error bound applicable to arbitrary time schedules, thereby offering insights into the design of schedules that minimize sample complexity. The schedule presented in Section 5 is designed based on this theorem, wherein we stipulate that $h_k$ should be proportional to $\bar{\gamma}_k^2$ to ensure a well-balanced discretization error.
>
> Specifically, for the case where $\gamma(t)=\sqrt{at(1-t)}$, the derived schedule manifests as an exponentially decaying schedule, characterized by a reduction in step size on both ends of the interval $[0,1]$. While this schedule shares similarities with the exponentially decaying schedule employed in diffusion models, a key distinction lies in the fact that diffusion models typically decay the step size on only one side of the interval.
>
> Furthermore, it is noteworthy that for alternative choices of $\gamma$, the optimal schedule may deviate from the above schedule.
>
> ### Experimental Designs Or Analyses
>
> > Although the authors provide theoretical justification for their method’s superiority, the experiments themselves appear overly simple. I recommend conducting toy experiments on low-resolution datasets such as CIFAR-10 or ImageNet32 since training flow models on these datasets are relatively fast and computationally manageable.
>
> A: Primarily, this work constitutes a theoretical investigation, wherein we establish the first discrete-time complexity analysis for the stochastic interpolants framework. Our contributions include the derivation of an explicit error bound for the discrete-time sampler, as presented in Theorem 4.3, and the subsequent exploration of schedule design strategies aimed at minimizing sample complexity. The objective of our numerical experiments is to validate our theoretical findings.
>
> While we acknowledge the value of demonstrating our method's efficacy on more complex datasets, such as CIFAR-10 or ImageNet32, we must emphasize that the application of our framework to these datasets introduces a multitude of practical challenges. These challenges encompass the design of effective network architectures, the meticulous tuning of hyperparameters, and the optimization of learning algorithms. Addressing these practical considerations falls outside the primary scope of this theoretical study. However, we recognize the importance of empirical validation on standard datasets and consider it an interesting direction for future research.
>
> > Given that the authors’ theory largely transfers discrete-time analysis to flow models, it would be more convincing to present additional experiments that demonstrate the claimed ability to “design efficient schedules for convergence acceleration.”
>
> A: A core contribution of our work lies in the derivation of the inaugural theoretical error bound for distribution estimation within the stochastic interpolants framework, explicitly expressed in terms of the step sizes. Leveraging this result, we theoretically demonstrate the feasibility of designing schedules that accelerate convergence by optimizing the balance between error terms.
>
> To empirically validate the efficacy of our schedule design methodology, we have conducted numerical experiments for both $\gamma=\sqrt{(1-t)t}$ (Section 6) and $\gamma=\sqrt{(1-t)^2t}$ (Appendix D.1), comparing the performance of our specifically designed schedules against that of the standard uniform schedule. While we acknowledge the potential for further experimentation to strengthen our claims, we consider the current numerical results as a solid validation. Furthermore, the exploration of optimal choices for $\gamma$ in practical applications represents an intriguing avenue for future research.
>
> ### Essential References Not Discussed
>
> A: Thank you for pointing them out! We will discuss them in the revision.
>
> ---
> We hope our response addresses your concerns. If so, we wonder if you could kindly consider raising your score? We will also be happy to answer any further questions you may have. Thank you very much!

---

> > ### Comment · Reviewer_s1b3 · 2025-04-07
> >
> > Thank you for clarifying your concerns. While I acknowledge the paper's theoretical contributions and the exploration of a flow-based model, I remain unconvinced that the proposed “stochastic interpolants” framework significantly differs from standard SDE-based diffusion models. In addition, the experimental results appear limited.
> >
> > Given the paper’s theoretical focus and the solid proofs provided, I recognize the authors’ contributions. However, these do not fully address my reservations regarding novelty and empirical support. Therefore, I maintain my initial score of 3.

---

### Official Review · Reviewer_5Shm · 2025-03-19

**Overall Recommendation:** 4

**Summary:**

This paper proposes a finite time analysis of discretization of stochastic interpolants. The paper presents assumptions on the initial and final distributions, score estimators, then provides a complexity bound in KL divergence.

**Claims And Evidence:**

The paper claims to approach the problem of analysis discrete-time stochastic interpolants, and does provide clear assumptions and results to support the claim. There are also numerics that demonstrate the convergence rate. No claims seem problematic.

**Essential References Not Discussed:**

None.

**Experimental Designs Or Analyses:**

The experimental design is very limited, limited to 2D examples. Given the explicit dimension dependence in the bound, I suggest authors to provide experiments with variable $d$ (maybe Gaussian case or something similar) and let $d$ grow, to obtain a scaling result to see whether their bound is predictive of what will happen with increasing $d$.

**Methods And Evaluation Criteria:**

Not applicable as this is mainly a theoretical paper.

**Other Comments Or Suggestions:**

1) line 149 lhs, "Both $b(t,x)$ and $b_F(t,x)$ can be expressed as linear combinations", clarify for reader's convenience.

2) Line 383, $N$ is the number of iterations, line 375 $N(0, I_d)$ is a Gaussian, use $\mathcal{N}$ for the density.

**Other Strengths And Weaknesses:**

Analysis seem very favourable dependence to dimension and $\varepsilon$ accuracy.

**Questions For Authors:**

1) The authors assume in Assumption 4.2 that the drift is estimated with high accuracy. However, can authors clarify how this can be relevant to practice? We know that $b_F$ can be decomposed as the score + velocity term. For the score term, very good estimators exist from diffusion models literature. Can the same be said for the second term? Is this a practical setting?

Please mention some practical estimators for the second term after this assumption, their behaviour, whether it is easy to ensure good training of it, as in the score case.

2) While the bound in Theorem 4.3 has a favourable dependence to dimension, I feel like some of this is hidden in the term $\mathbb{E}\|x_0 - x_1\|^8$. Please exemplify a bound with two multivariate Gaussians - provide this expression and provide its dependence to $d$. Similar question extends to Corollary 5.2.

4) What is the reason in Assumption 4.1 that you have an 8th moment assumptions, vs. 4th moments in the literature. Please give some intuition following line 190.

5) Figure 1: is there any issue with $t = 0.001$. Is the estimator $\hat{b}_F$ stable around this region? Please provide the plot around this time for the SDE.

6) I suggest authors to not assume uniformly that $b_F$ is estimated $\varepsilon$-accurate. Perhaps given the score estimators, you can provide $\varepsilon_{score}$-accurate score and $\varepsilon_{vel}$-accurate velocity term estimate - thus see the interplay between these two error sources and how they play out in your final bound.

**Relation To Broader Scientific Literature:**

The paper extends the work on stochastic interpolants, particularly Albergo et al. (2023) and provides a discrete time analysis. However, my opinion in general is that given the assumption on $\varepsilon$-accurate drift for the SDE given in the Albergo et al. (2023), I think the contribution is a bit incremental - this is a discretisation analysis in essence. Which is usually standard adapting the usual analysis methods as authors did.

**Theoretical Claims:**

While I did give a careful look at statements and proof sketches, I did not check every single line of proofs. Below are my high-level questions about theoretical claims

---

> ### Author Rebuttal · Authors · 2025-03-31
>
> Thank you for your time and effort in reviewing our paper! We are grateful for your constructive suggestions, which have significantly guided our improvements. Please find our responses to your comments below.
>
> ### Experimental Designs Or Analyses
>
> A: Thanks for the suggestion. We tested the samplers on $d$-dimensional Gaussian mixtures ($\rho_0$ and $\rho_1$) and compared the distribution estimation error for different $d$. The real drift terms used in the sampler are analytically computed [1]. You can check [link1](https://drive.google.com/file/d/1PuZy-aR9OYwBtB9xdnPaxn5qAP2G3-3O/view) or [link2](https://drive.google.com/file/d/1wkOsPsqq0wzCCd8paGZL8qIpK2aYMD52/view) for results. The first figure compares the error for different $d$ with fixed iterations, and the second shows convergence for higher dimensions. Both results align with our theory and validate the $d$ dependence.
>
> ### Other Comments Or Suggestions
>
> A: Thank you for your suggestions! We will update them in the revision.
>
> ### Questions For Authors
>
> A1: We first emphasize that our work primarily focuses on the theoretical analysis of the general stochastic interpolant method. We provide the first discrete-time analysis and propose an upper bound for the distribution estimation error. Additionally, we develop a discretization time schedule that optimizes this bound for a specified $\gamma$.
>
> Theorem 4.3 offers a general upper bound, controlling the error using $\varepsilon_{b_F}^2$, the distance between distributions, data dimension $d$, and step sizes $h_k$. Assumption 4.2 quantifies how close $\hat b_F(t,x)$ is to $b_F(t,x)$ and holds for some $\varepsilon_{b_F}^2$ if $\hat b(t,x_t)$ has finite second moments. Theorem 4.3 illustrates how this impacts the final distribution error.
>
> Our numerical experiments validate these findings. We use the loss function $\mathcal{L}[\hat{v}]=\frac{1}{2}\int_{t_0}^{t_N}\mathbb{E}[|\hat{v}|^2-2\partial_tI\cdot\hat{v}]\text{d}t$ to train the estimator $\hat{v}(t,x)$ for $v(t,x)$. This choice proved effective and is commonly used (e.g., [1], [2], [3]).
>
> However, creating robust estimators for general applications is challenging, requiring careful designs for network architectures, learning algorithms, and learning rate schedules, which is a direction for future research.
>
> A2: Regarding dimension dependence in Theorem 4.3, the key terms contributing to the discretization error are:
> $$\varepsilon_{\text{dis}}\lesssim\sum_{k=0}^{N-1}h_k^3(d^3\bar \gamma_k^{-6}+d\sqrt{\mathbb{E}|x_0-x_1|^8}\bar\gamma_k^{-2})+\sum_{k=0}^{N-1}h_k^2(d^2\bar\gamma_k^{-4}+d\sqrt{\mathbb{E}|x_0-x_1|^4}\bar\gamma_k^{-2}).$$
> For a multivariate Gaussian $z\sim\mathcal{N}(0,I_d)$, $\mathbb{E}|z|^{2p}\le C(p)d^p$. If $x_0$ and $x_1$ are multivariate Gaussians, $\mathbb{E}|x_0-x_1|^8=O(d^4)$. Substituting this shows the dependence on $d$ is $O(d^3)$ in the $h_k^3$ term and $O(d^2)$ in the $h_k^2$ term. The dimensional dependence is also consistent in Corollary 5.2.
>
> A3:Our work is focused on discrete-time analysis, which is new for the stochastic interpolant framework. Unlike continuous-time error bounds that use the 4th moment assumption (see [1]), our discrete-time analysis requires the 8th moment assumption. In Theorem 4.2, we control $\mathbb{E}|\nabla b_F(t,x_t)\cdot b_F(t,x_t)|^2$, requiring $\mathbb{E}|x_0-x_1|^8<\infty$. This term appears as $h_k^3\bar{\gamma}_k^{-2}d\sqrt{\mathbb{E}|x_0-x_1|^8}$ in Theorem 4.3's error bound.
>
> However, this term doesn't dominate when the step size is small, with the error bound being dominated by $h_k^2\bar{\gamma}_k^{-2}d\sqrt{\mathbb{E}|x_0-x_1|^4}$. Relaxing the 8th moment assumption is a future research direction.
>
> A4: It is true that the drift term $b_F(t,x)$ has larger variation near boundaries, motivating our exponentially decaying schedule in both theory and experiments. Our estimator approximates $b_F(t,x)$ well for $t_0=0.001$, so this choice does not pose a problem. We will add more details and visualizations in the revision.
>
> A5: While analyzing the estimation errors of score and velocity terms separately is suggested, we find it unnecessary for our analysis. Partitioning $b_F(t,x)=v(t,x)+(\epsilon-\dot{\gamma}\gamma)s(t,x)$ results in $\varepsilon_{b_F}^2\lesssim\varepsilon_v^2+\sup(\epsilon-\dot{\gamma}\gamma)^2\cdot\varepsilon_s^2$, which is similar to [1] for continuous-time stochastic interpolants.
>
> ### References
>
> [1] Albergo, M. S., Boffi, N. M., and Vanden-Eijnden, E. Stochastic interpolants: A unifying framework for flows and diffusions, 2023.
>
> [2] Yaron Lipman, Ricky T. Q. Chen, Heli Ben-Hamu, Maximilian Nickel, and Matthew Le. Flow matching for generative modeling, 2023.
>
> [3] Albergo, M. S. and Vanden-Eijnden, E. Building normalizing flows with stochastic interpolants, 2023.
>
> ---
> We hope our response addresses your concerns. If so, we wonder if you could kindly consider raising your score? We will also be happy to answer any further questions you may have. Thank you very much!

---

> > ### Comment · Reviewer_5Shm · 2025-04-06
> >
> > _I posted this comment yesterday as an 'official comment' without realizing that this is apparently not visible to authors. Please read below my response:_
> >
> > Thank you!
> >
> > Thank you for the plots. Can you elaborate how one would interpret your linear growth plot in link1 in the context of Theorem 4.3? Does it connect with the theoretical result? Also, if you were to fix some
> >  and run numerics with increasing , scaling the number of steps necessary for this, would you get the same scaling as in theory?
> >
> > I am happy to raise my score, if these experiments were conducted with good care. To be honest, I don't expect perhaps the behaviour in these experiments to match theory, as theory is just an upper bound. But this would give a quite good picture whether there is more work to be done, or bounds derived here are tight.

---

> > > ### Author Response · Authors · 2025-04-07
> > >
> > > Thank you for your comment! We are glad to further discuss the questions with you. Please see our response below.
> > >
> > > In our experiments, the KL error scales almost linearly w.r.t. $d$. The $O(d^2)$ KL error bound provided by our theorem holds for this case, and the linear growth of KL error in the experiments can be caused by several reasons. For example, there might be some special properties of gaussian mixtures which does not hold for general distributions, or there simply exists a sharper bound that haven't been found. Trying to improve the bound would be an interesting work in the future.
> > >
> > > In addition, we have scaled the number of iterations ($N$) in our experiments on Gaussian mixture data, which makes the convergence clearer, see [link3](https://drive.google.com/file/d/1YIIf0KlcrXrw1u1ybS-KDya_-twRKle0/view?usp=drive_link) and [link4](https://drive.google.com/file/d/1_7aaaSrkrgYOKl6KM1plM1scW3S0nfzm/view). Moreover, compared to [link1](https://drive.google.com/file/d/1PuZy-aR9OYwBtB9xdnPaxn5qAP2G3-3O/view), we draw multiple curves in [link4](https://drive.google.com/file/d/1_7aaaSrkrgYOKl6KM1plM1scW3S0nfzm/view) to show the scale of KL error w.r.t. the dimension $d$ for fixed number of iterations.
> > >
> > > ---
> > > We hope our response addresses your concerns. If so, we wonder if you could kindly consider raising your score? We will also be happy to answer any further questions you may have. Thank you very much!

---

### Official Review · Reviewer_9oGn · 2025-03-21

**Overall Recommendation:** 4

**Summary:**

This paper derives a convergence bound for the stochastic interpolants framework. The discretized SDE analyzed in this work is more general than the SDE analyzed in existing diffusion model convergence bounds. Like in the diffusion setting, the derived bound suggest an that an exponentially decaying discretization schedule is a better choice. The authors show empirically that this is indeed the case.

## update after rebuttal

I thank the authors for their clarifications. The more general interpolant analysis remains quite close to existing work but I believe it could be interesting to have the result out there, I therefore increase my score.

**Claims And Evidence:**

The central claim in the paper is the convergence bound is provided in Theorem 4.3. The proof adapts the result of Chen et al [A] on the discretization of SDE to the stochastic interpolant framework. The differences introduced by the slightly more general interpolant I(x_0, x_1, t) are dealt with in Lemmas B.3 and C.1, which are the main technical contribution of the work (Could the authors confirm?).

A secondary claim is derived from the theorem which shows that an exponentially decaying schedule cancels out terms appropriately and leads to a simplified bound in Proposition 5.1. This claim is confirmed experimentally in Figure 3.

---
[A] Chen, Sitan, et al. "Sampling is as easy as learning the score: theory for diffusion models with minimal data assumptions."

**Essential References Not Discussed:**

.

**Experimental Designs Or Analyses:**

.

**Methods And Evaluation Criteria:**

.

**Other Comments Or Suggestions:**

.

**Other Strengths And Weaknesses:**

The strengths: The paper is clearly written and easy to follow. It adapts prior work and adds to the literature on discretized SDEs used for sampling. The derived bound gives indication on a good choice of step-size schedule and the authors confirm this experimentally.

A minor weakness: The technical extension over prior work might be slightly limited but it can still be worthwhile to have the bound in the literature.

**Questions For Authors:**

Could an additional assumption on the target distribution remove the unfortunate initial KL term in the bound?

**Relation To Broader Scientific Literature:**

There are some papers already providing analysis of the flow matching framework and the authors clearly compare their techniques with prior work.

**Theoretical Claims:**

The analysis is sound. The only unfortunate element is the need for $t_0 > 0$ and the initialization error resulting from this. It creates a discrepancy between the bounds in the paper and bounds for diffusion models whose initialization error can be made exponentially small by increasing the simulation interval T.

---

> ### Author Rebuttal · Authors · 2025-03-31
>
> Thank you for your time and effort in reviewing our paper! We are grateful for your constructive suggestions, which have significantly guided our improvements. Please find our responses to your comments below.
>
> ### Claims And Evidence / Weakness
>
> > The differences introduced by the slightly more general interpolant $I(x_0, x_1, t)$ are dealt with in Lemmas B.3 and C.1, which are the main technical contribution of the work (Could the authors confirm?).
> > A minor weakness: The technical extension over prior work might be slightly limited but it can still be worthwhile to have the bound in the literature
>
> A: The main contributions of our work include the following. First, we propose the first discrete-time analysis for the stochastic interpolant framework, where we propose a new discrete-time sampler and provide the rigorous distribution estimation error bound for the sampler (Theorem 4.3). Second, based on the error bound, we further develop a time schedule for the discretization to achieve lower sample complexity (Corollary 5.2), when the latent scale is defined by $\gamma(t)=\sqrt{at(1-t)}$. Lastly, we validate our theoretical results with numerical experiments.
>
> Among our technical results, Lemma B.3, Lemma C.1 and Appendix B.2 address the discretization error of the drift term $b_F(t,x)$. In this part, the main difference between our result and previous results is that we utilize a more general interpolant $I(t,x_0,x_1)+\gamma(t)z$ between two general distributions. This introduces a novel velocity term $v(t,x)$ in the process, and demands new analysis for the estimation error compared to existing results on diffusion models.
>
> ### Questions
>
> > Could an additional assumption on the target distribution remove the unfortunate initial KL term in the bound?
>
> A: Firstly, the initial KL term quantifies the discrepancy between the true initial distribution $\rho(t_0)$ and the estimated distribution $\hat{\rho}(t_0)$, where $\hat X_{t_0}$ is sampled from. We incorporate this term into our theorem to comprehensively cover the scenario where $\hat{X}_{t_0}$ differs from $\hat{\rho}(t_0)$, and provide a corresponding error bound in Theorem 4.3
>
> Next, we look at two examples on how this term affects our error bound. If we define $I$ such that $I(t,x_0,x_1)=x_0$ within the interval $t\in[0,t_0]$, the initial error becomes exactly $0$ because we can directly sample from $\rho(t_0)$ (note that during generation, data from $\rho_0$ is accessible). Furthermore, if $\gamma^2(t)=\Theta(t)$ near $t=0$, we can achieve an $O(t_0)$ initial KL error if we choose $\hat{\rho}(t_0)$ such that $\hat{X}_{t_0}=x_0+\gamma(t)z$ for $x_0\sim\rho_0$ and $z\sim\mathcal{N}(0,I_d)$. In this case, the initial KL error is the same to the initial KL error of diffusion models with $T=\Theta(\log(1/t_0))$ [1].
>
> ### References
>
> [1] Benton, J., Bortoli, V. D., Doucet, A., and Deligiannidis, G. Nearly d-linear convergence bounds for diffusion models via stochastic localization
>
> ---
> We hope our response addresses your concerns. If so, we wonder if you could kindly consider raising your score? We will also be happy to answer any further questions you may have. Thank you very much!

---

### Decision · Program_Chairs · 2025-05-01

**Decision:**

Accept (poster)

**Comment:**

This paper conducts a convergence analsysis of discretization of finite time stochastic interpolants.
The authors further demonstrate how their theoretical results have practical implications on time-step scheduling.
All reviewers appear to agree that the paper is theoretically sound.
However, two reviewers raises conerns regarding the novelty of the work in relation to diffusion/flow models.
All reviewers appear to agree that the experimental results are sufficient to demonstrate the theoretical contribution.
However, one reviewer wanted empirical exploration beyond the theory; in particular investigation into the established bound as the dimension increases.
Another reviewer wanted more empirical validation "in the wild" such as CIFAR-10 or ImageNet32.

In a camera-ready verison I would suggest that the authors consider:

1. More clearly explain what the challenges are in transferring previous result on discretization of diffusion models to the stochastic interpolant framework.
2. Empirical investigation of the error bound in relation to the dimension.